# Projections and uncertainties of winter windstorm damage in Europe in a changing climate

Luca G. Severino[1], Chahan M. Kropf[2,3], Hilla Afargan-Gerstman[1], Christopher Fairless[2], Andries Jan de Vries[4], Daniela I.V. Domeisen[4,1], and David N. Bresch[2,3]

[1]Institute for Atmospheric and Climate Science, ETH Zürich, Zürich, Switzerland
[2]Institute for Environmental Decisions, ETH Zürich, Zürich, Switzerland
[3]Federal Office of Meteorology and Climatology MeteoSwiss, Zürich, Switzerland
[4]Institute of Earth Surface Dynamics, University of Lausanne, Lausanne, Switzerland

**Correspondence:** Luca Severino (luca.severino@usys.ethz.ch)

**Abstract.**

Winter windstorms are among the most significant natural hazards in Europe linked to fatalities and substantial economic damages. However, projections of windstorm impact in Europe under climate change are highly uncertain. This study combines climate projections from 30 general circulation models participating in CMIP6 with the climate-risk assessment model CLIMADA to obtain projections of windstorm-induced damages over Europe in a changing climate. We conduct an uncertainty-sensitivity analysis, and find large uncertainties in the projected changes in the damages, with climate model uncertainty being the dominant factor of uncertainty in the projections. We investigate spatial patterns of the climate change-induced changes in windstorm damages and find an increase in the damages in northwestern and northern-central Europe, and a decrease over the rest of Europe, in agreement with an eastward extension of the North Atlantic storm track into Europe. We combine all 30 available climate models in an ensemble of opportunity approach and find evidence for an intensification of future-climate windstorm damages, with damages with return periods of 100 years under current climate conditions becoming damages with return periods of 28 years under future SSP585 climate scenarios. Our findings demonstrate the importance of climate model uncertainty for the CMIP6 projections of windstorms in Europe, and emphasize the increasing need for risk mitigation due to extreme weather in the future.

## 1 Introduction

Extratropical cyclones (ETCs) can cause intense windstorms in winter (hereafter referred to as *winter storms*), and are among the most significant natural hazards in Europe in terms of fatalities and damages to physical assets (Schwierz et al., 2010). Understanding future changes in winter storm risk is thus key for risk assessment and damage mitigation in Europe.

Both the main regions of extratropical cyclonic activity (known as the storm tracks) and the characteristics of individual ETCs are expected to change as climate changes (Shaw et al., 2016; Catto et al., 2019). Many studies suggest an eastward extension of the North Atlantic storm track further into Europe, potentially increasing the risk of winter storms and their regional impacts (Zappa et al., 2013; Zappa and Shepherd, 2017; Oudar et al., 2020; Harvey et al., 2020; Priestley and Catto, 2022).

However, changes in the North Atlantic storm track are highly uncertain, as climate models feature biases in both the location and the dynamical intensity of the storm tracks (Lee et al., 2021). For instance, General Circulation Models (GCMs) participating in the Coupled Model Intercomparison Project (CMIP) are associated with a winter storm track extending too zonally into Europe (Harvey et al., 2020; Priestley et al., 2020), and tend to underestimate the intensity of the most intense ETCs (Seiler and Zwiers, 2016; Priestley et al., 2020). In addition, the level of global warming will likely be an important factor of uncertainty, with emission scenarios corresponding to higher radiative forcings potentially further increasing both the number and the intensity of ETCs associated with strong wind speeds over northwestern Europe (Zappa et al., 2013; Priestley and Catto, 2022).

Modelling the impacts of winter storms and their strong winds can be achieved by incorporating surface wind projections from climate models into climate-risk models, thus using the risk framework to derive estimates of natural hazard-related impacts on natural or socio-economic systems. In the risk framework, risk from a natural hazard can be modelled as the convolution between three components: the hazard, the exposure, and the vulnerability (IPCC, 2014). The hazard describes the distribution of the intensity of a certain hazard in space and time (e.g. wind gust intensities, flooding heights), the exposure describes the distribution of assets at risk in space and time (e.g. population, infrastructure, crops, ecosystems), and the vulnerability makes the link between hazard and exposure, by describing the effect of the hazard on the exposed value in terms of the assessed impact. Most often, the complex hazard-exposure-impact relation of the vulnerability component is approximated with simpler functional relationships, called *vulnerability curves*, or *impact functions*.

By incorporating the three components of risk, climate-risk models can jointly assess changes in exposure and vulnerability and changes in the climate system, thus allowing to produce comprehensive estimates of climate impacts. These estimates of climate impacts are extremely valuable for the mitigation and management of changing climate risks, and can greatly help to improve the resilience of our societies or of the natural environment to climate change.

Considerable uncertainties are involved in the modelling of the hazard, exposure and vulnerability, which in turn can contribute to render risk projections highly uncertain. Understanding and quantifying those uncertainties, and their influence on the outcome of the risk projection is crucial for risk assessment studies. Uncertainty and sensitivity analyses are a commonly used method to study and model uncertainties associated with the use of complex models such as climate-risk models (Saltelli et al., 2008; Pianosi et al., 2016; Kropf et al., 2022). Uncertainty and sensitivity analyses rely on the generation of many replications of a model's output, where the replications are obtained by varying some of the model's input factors which are considered as uncertain. The term *input factors* here refers to any of the model's components (e.g. parameters, data) which can drive a variation in any of the model's outputs, while the term *outputs* refers to any numerical results produced by the model (Saltelli et al., 2008, 2019). Uncertainty analysis can then be understood as the study of the distribution of the generated model outputs, and sensitivity analysis as the study of the links between the model's uncertain input factors and the generated model outputs. Uncertainty and sensitivity analyses can thus provide insights into which of the model's input factors contribute most to the uncertainty in model output, and help in understanding the model's input-output relationships.

Many recent studies have focused on impacts of extratropical winter windstorms in Europe under past (e.g. Della-Marta et al., 2010; Donat et al., 2010b; Haylock, 2011; Priestley et al., 2018; Vautard et al., 2019; Walz and Leckebusch, 2019; Koks and Haer, 2020; Welker et al., 2021; Röösli et al., 2021; Wilkinson et al., 2022) or future (e.g. Leckebusch et al., 2007; Heneka et al., 2007; Pinto et al., 2007; Schwierz et al., 2010; Donat et al., 2011; Pinto et al., 2012; Ranson et al., 2014; Karremann et al., 2014; Hochman et al., 2022; Little et al., 2023) climate conditions. For instance, Pinto et al. (2007) assessed future European winter storm damages considering multiple realizations of a single climate model with varying initial conditions and boundary forcing scenarios, and Leckebusch et al. (2007) and Donat et al. (2011) assessed future European winter storm damages considering different climate models. However, only a limited number of studies made use of the most recent projections from climate models participating in phase six of the Coupled Model Intercomparison Project (CMIP6; Eyring et al., 2016) to assess winter storm related wind damages in Europe. Notably, Little et al. (2023) used an ensemble of eight climate models participating in CMIP6 to conduct a multi-model and multi-scenario assessment of winter storm risk over Europe. Furthermore, a few uncertainty and sensitivity analyses focusing on winter storm damage risk modelling under current climate have been published (e.g. Koks and Haer, 2020; Röösli et al., 2021), but a comprehensive estimate of the importance of the different uncertainties involved in the modelling of winter storm damage in Europe under future climate is still lacking.

This study combines state-of-the-art climatic projections from 30 global climate models participating in CMIP6 with the open-source weather and climate risk assessment model CLIMADA to obtain a set of relevant projections for winter-storm-induced wind damages to physical assets over Europe in a changing climate. CLIMADA's uncertainty-sensitivity quantification module *unsequa* (Kropf et al., 2022) is used to quantify the importance of five different sources of uncertainty in the projections: Climate model, internal climate variability, future climate scenario, exposure, and impact function uncertainty.

This paper is structured as follows: Sect. 2 introduces hazard, exposure, and vulnerability data, and presents the damage and uncertainty modelling approaches; Sect. 3 presents the results of both the uncertainty and sensitivity quantification, and the winter storm damage projections; Sect. 4 summarizes, discusses the results, and concludes.

## 2 Data and methods

This study uses the open-source weather and climate risk modelling platform CLIMADA (Aznar-Siguan and Bresch, 2019) to estimate windstorm damages associated with European winter storms in a changing climate. The term damage here represents the economic losses resulting from the impact of intense surface wind gusts on physical assets. CLIMADA is based on the risk framework as defined by the IPCC (IPCC, 2014) that incorporates the three components hazard, exposure, and vulnerability to compute impact and risk metrics in a fully probabilistic and geographically explicit manner. Additionally, CLIMADA features an uncertainty and sensitivity quantification module (*unsequa*; Kropf et al., 2022), which is used in this study to assess the key uncertainties associated with the modelling of the damages. We study the impact of climate change on winter storm damage over Europe, by combining hazard data modelled for a domain comprised within latitudes 30° to 75°N and longitudes 30°W

to 30°E with exposure data for 44 European countries. To better explore the spatial patterns of the damages, specific countries are grouped into seven sub-regions shown in Fig. 2, the British Isles (BI), the Iberian Peninsula (IP), Western Europe (WEU), Central Europe (CEU), the Mediterranean and Balkan region (MED), Scandinavia (SC), and Eastern Europe (EEU). The seven sub-regions are defined following their climate and their exposure to winter storm hazards, following Christensen and Christensen (2007). Damages are investigated on a daily basis, for the winter half-year only (October-November-December-January-February-March). Climate change effects are studied by comparing damages computed for a future (2070-2100) versus a historical (1980-2010) period while keeping exposure and vulnerability invariant in time. We present our results as the difference between the damages computed for the future and the damages computed for the historical reference period, divided by the damages computed for the historical reference period. We call this approach *Delta Climate*, as it informs on the change in winter storm damage associated with changing climate conditions but disregarding future changes in exposure and vulnerability. GCMs participating in CMIP6 are used to represent past and future climates and their surface winds as described in Sect. 2.1. Exposure and vulnerability data are described in Sect. 2.2 and 2.3. Section 2.4 describes the different damage and risk metrics used in this study, and Sect. 2.5 briefly details the methods and results of the model's calibration and validation procedures. Finally, Sect. 2.6 introduces the *unsequa* module and explains the uncertainty and sensitivity quantification framework. See Fig. 1 for a visual summary of the study framework.

## 2.1 CMIP6 windstorm hazard data

We use daily surface wind maximum (sfcWindmax) outputs from 30 GCMs participating in CMIP6 to represent winter storm hazards. Model data is kept on the original model grids, as provided by the modelling centers. We select the GCMs on the criterion that a GCM provides at least one simulation for the historical period and one simulation for the future period, obtained using the forcing dataset from the Shared Socio-economic Pathway 5-8.5 (hereafter SSP585; for a detailed description of the shared socio-economic pathways and CMIP6 scenarioMIP experiments, see O'Neill et al., 2016). We choose the high-emission SSP585 scenario as we expect it to correspond to a high-impact scenario (Zappa et al., 2013). For each GCM, a maximum of three ensemble members is considered. One ensemble member of the historical and one ensemble member of the SSP585 simulations from the GCM NESM3 were discarded from the analysis due to a strong negative bias in the surface wind speeds of the historical member, which resulted in a strong positive bias in future-minus-historical change in surface wind maxima. Table A1 summarizes the climate models and climate model members used for the damage projections of this study. Model members representing the same GCM-SSP combination are assembled into one single simulation in order to decrease the effects of internal variability. Thus, for each of the study periods (historical or future), we obtain hazard data with a duration of 30, 60, or 90 years, for climate models with respectively one, two, or three ensemble members. First, we consider each climate model separately to compute the damage projections (Sect. 3.2). Considering the climate models separately allows us to investigate the climate model uncertainty in the projections. As a second step, we combine the different climate models, in an ensemble of opportunity approach (Tebaldi and Knutti, 2007), where all models are considered to be equally valid realizations of the climate and combined without prior weighting (Sect. 3.3). This ensemble of opportunity approach allows us to consider simulations representing 400 winter seasons, which can be used to estimate damage events with considerably longer return

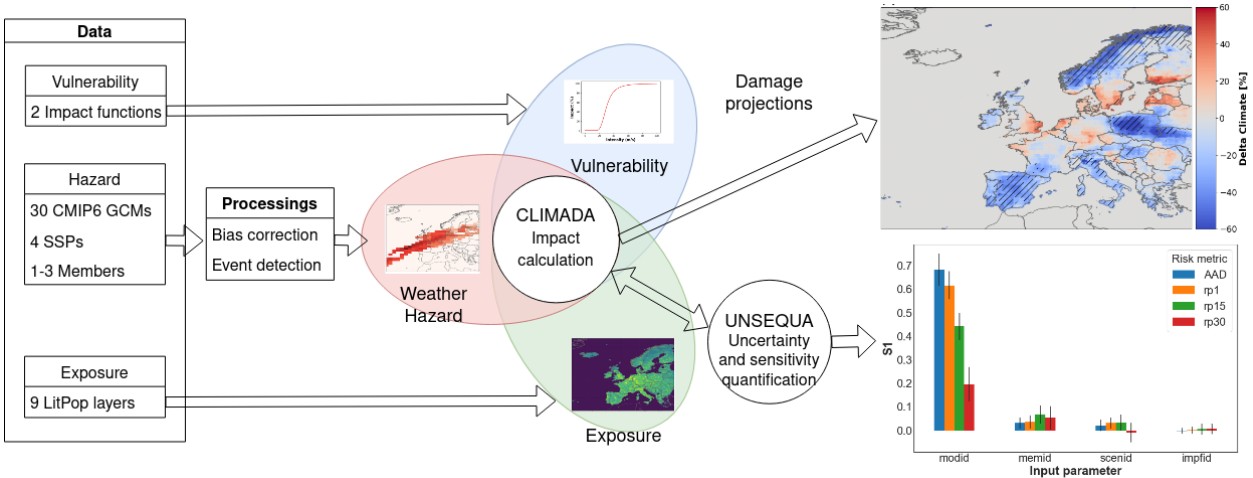

**Figure 1.** Schematic describing the modelling process. Thirty General Circulation Models (GCMs) participating in phase six of the Coupled Model Intercomparison Project (CMIP6) and featuring one to three ensemble members are used to represent the winter climate from a historical (1980-2010), and a future (2070-2100) period, resulting in 67 members x 30 years x 6 months = 12060 months of winter climate for each period. The historical climates are obtained from CMIP6's historical experiments and the future climates from the Shared Socio-economic Pathway 5-8.5 (SSP585) experiments. Daily surface wind maximum outputs from each GCM are bias-corrected using a percentile mapping approach, and storm days are detected from the pre-processed daily surface wind maximum fields from those 2 x 12060 months of winter climate. The storm days derived from the GCMs, exposure data from the Litpop dataset, and a storm damage impact function taken from Schwierz et al. (2010) are then incorporated as the weather hazard, exposure, and vulnerability data into the weather and climate risk assessment model CLIMADA, which then produces damages and risk metrics, such as damage maps, or Exceedance-Frequency-Curves. We use those damage and risk metrics to obtain *Delta Climate* estimates, where *Delta Climate* refers to the future-minus-historical change in the metric relative to the historical period. As a second step, CLIMADA's uncertainty and sensitivity quantification module (*unsequa*) is used to study the uncertainty and sensitivity related to the hazard, exposure, and vulnerability components in the damage projections. To this end, we generate additional hazard data from 14 GCMs featuring three realizations for each of the historical, SSP126, SSP245, SSP370, and SSP585 experiments from CMIP6; eight additional exposure data layers using different parameterizations of LitPop; and an additional storm damage impact function taken from Klawa and Ulbrich (2003). This additional hazard, exposure, and vulnerability data is then used by CLIMADA's *unsequa* module to quantify the uncertainty in several damage and risk metrics, including the Average Annual Damage (AAD), and damage amounts with return periods, of one, 15, and 30 years (rp1, rp15, rp30), and to assess the sensitivity of those damage and risk metrics to different components of the modelling framework, including $modid$: climate model choice; $memid$: climate model member choice; $scenid$: future climate scenario choice; and $impfid$: impact function choice.

periods.

We use a percentile mapping technique originally designed in Rajczak et al. (2016) and adapted by Lüthi et al. (2022) to bias-correct the GCMs' daily sfcWindmax outputs. ERA5 10m wind gust maxima (WG10) are used as reference for the bias correction. Hourly WG10 data are first resampled to a daily resolution and then linearly interpolated to each CMIP6 GCM's

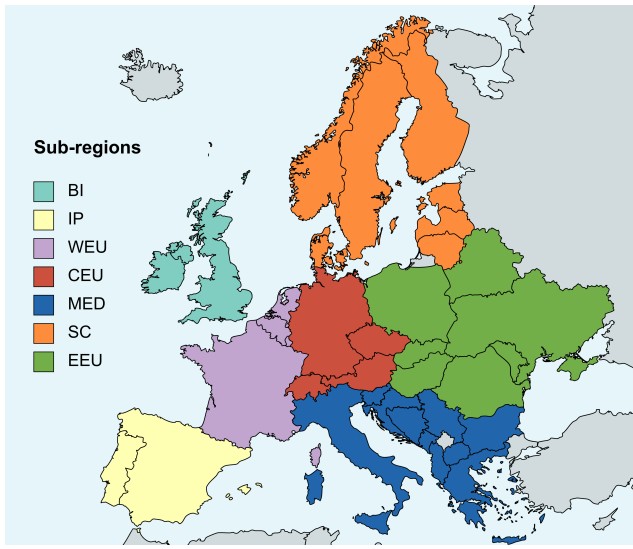

**Figure 2.** Map of the seven sub-regions used for regional winter storm damage assessment: the British Isles (BI; United Kingdom, Ireland), the Iberian Peninsula (IP; Spain, Portugal, Andorra), Western Europe (WEU; France, Monaco, Kingdom of the Netherlands, Luxembourg, Belgium), Central Europe (CEU; Switzerland, Germany, Liechtenstein, Czech Republic, Austria), the Mediterranean and Balkan region (MED; Italy, Albania, Bosnia and Herzegovina, Croatia, Montenegro, Malta, Greece, San Marino, Vatican City State, Slovenia, Macedonia, Bulgaria, Serbia), Scandinavia (SC; Denmark, Sweden, Finland, Norway, Estonia, Latvia, Lithuania), and Eastern Europe (EEU; Belarus, Hungary, Poland, Romania, Slovakia, Ukraine, Republic of Moldova). Physical asset exposure is modelled for each of the listed countries using the LitPop dataset (Sect. 2.2).

grid. A bias-correction is then carried out for each climate model's grid-cell, using one single correction function per GCM,
so that inter-member variability is preserved. For each climate model, the correction function is computed as the average correction function taken over the different ensemble members of the climate model.

We detect stormy days associated with European winter storms from the bias-corrected daily GCM data by applying a simple selection procedure, based on local wind statistics. A storm event is defined as a day for which stormy conditions are locally detected over a part of the domain. First, a grid-cell of a GCM's grid is considered as stormy if the daily intensity of the
sfcWindmax is in excess of its local 98th percentile value, computed over the winter half-years of the historical simulation of the model. The choice of the 98th percentile value as a threshold has been widely used for winter storm damage assessment studies (e.g., Klawa and Ulbrich, 2003; Pinto et al., 2007; Schwierz et al., 2010; Donat et al., 2010a, 2011), and is based on the assumption that storms and associated damages only occur during the two percent windiest days of the winter half-year (Klawa and Ulbrich, 2003). This modelling approach does not assume any temporal adaptation to future changes in the wind climate,
as the local wind speeds thresholds are constant in time and representative of the historical wind conditions only. We also ensure that the selected daily wind fields correspond to wind intensities that are sufficiently intense to produce actual damages by further requiring the daily wind intensities to be greater than a value of $15 \, \mathrm{m \cdot s^{-1}}$ to be considered as stormy (Schwierz

et al., 2010). Only grid-cells that fulfill both conditions are considered for the subsequent damage calculations. Secondly, the total area of the stormy grid-cells on a particular day must amount to a minimum threshold $A_{min}$ for the day to be considered

as stormy. Stormy grid-cells do not have to be contiguous to be included in the total stormy area required for a storm day. Hence, different wind features at separate geographical locations can be combined to evaluate whether or not the storm area exceeds the minimum threshold to count as a storm day in Europe. A value of 150000 km$^2$ is chosen for $A_{min}$, which is representative of the typical area of the wind footprint of an extratropical storm (Kruschke, 2014). Days that do not fulfill this minimum area requirement are not considered in the analysis, thus noise and small-scale events unrelated to European winter

storms are filtered out. In summary, a stormy day is defined as:

$$\text{Stormy day}_t \iff \sum_i \{a_i | [(v_{i,t} \geq v_{i,98}) \ \& \ (v_{i,t} \geq 15)]\} \geq A_{min} \tag{1}$$

where $a_i$ is the area of the grid cell $i$, $v_{i,t}$ is the daily sfcWindmax intensity at grid cell $i$ on the considered day $t$, $v_{i,98}$ is the 98th percentile of the daily sfcWindmax at grid cell $i$, computed over the winter periods of the historical period, and $A_{min}$ is the area threshold parameter.

## 2.2 LitPop exposure data

We use the LitPop dataset to represent exposure (Eberenz et al., 2020). LitPop provides geographical distribution of physical asset exposure, by spatially disaggregating country-specific macroeconomic indicators (e.g. produced capital, gross domestic product) using nightlight intensity and population count data. For this study we choose produced capital as the macroeconomic indicator, which we distribute in space at a resolution of 600 arcseconds and using the base parameterization of LitPop, which

gives equal weight to the nightlight and population disaggregation layers. We assume no future change in population and economy, and thus keep the exposed values and their geographical distribution constant in time. Keeping a time-invariant exposure baseline allows us to focus on the climate change impacts on the risk outcomes and is a common approach taken in the field of natural hazard risk modelling (e.g. Leckebusch et al., 2007; Pinto et al., 2007; Schwierz et al., 2010; Donat et al., 2011; Pinto et al., 2012; Karremann et al., 2014; Meiler et al., 2023; Rana et al., 2022; Stalhandske et al., 2022). However, the

assumption of a constant exposure over time yields an incomplete view of the future risk associated with storm damage, as exposure is expected to undergo considerable change over time, due to economic and societal development (IPCC, 2014). For a more complete view of future risk, changes in exposure over time should be accounted for (see e.g. Cremen et al., 2022).

## 2.3 Vulnerability

CLIMADA uses impact functions to represent vulnerability. Those impact functions link input wind gusts intensity to a mean

damage degree (MDD), representing the percentage of the exposure asset value damaged at the given wind intensity. Thus, each daily hazard value at each of the hazard data grid-cells is converted into a MDD, which CLIMADA then combines to the LitPop exposure data to compute damages at the exposure level. As a first step, we derive projections of winter windstorm damage under historical and future climate conditions using one impact function, which has been designed in Schwierz et al. (2010, hereafter Sw2010), and which is already implemented in CLIMADA. This function has been directly derived from

an insurer's loss model and is based on past claim data in the United Kingdom and was cross-validated with other European countries.

As a second step, we test the sensitivity of our results by considering a different impact function, which uses wind speed values above a threshold instead of absolute wind intensity. This excess-over-threshold impact function (hereafter CubEOT) relies on the assumption that damages only occur when wind is in excess of a local threshold, computed at the grid-cell level, and that

the damages are proportional to the cube of the excess-over-threshold intensity. This cubic excess-over-threshold relation has first been derived by Klawa and Ulbrich (2003), and then further used in a number of studies modelling winter storm damage in Europe (e.g., Pinto et al., 2007; Leckebusch et al., 2007; Donat et al., 2010b, 2011; Pinto et al., 2012). Consistent with our event definition, we use as a local threshold value the 98th percentile of the daily surface wind maximum computed over the historical period. We also give an upper limit to the damages potentially achievable, by assuming a constant MDD when wind

intensities are more than double their 98th percentile value. The function CubEOT is defined as:

$$MDD_{i,t} = \begin{cases} MDD_{max} & \text{if } v_{i,t} > 2 \cdot v_{98,i} \\ MDD_{max} \cdot \left( \frac{v_{i,t}}{v_{98,i}} - 1 \right)^3 & \text{if } 2 \cdot v_{98,i} \geq v_{i,t} \geq v_{98,i} \\ 0 & \text{otherwise} \end{cases} \tag{2}$$

where $MDD_{i,t}$ is the MDD computed for the day $t$ and model grid cell $i$, $MDD_{max}$ is a constant corresponding to the maximum MDD achievable and determined during calibration, $v_{i,t}$ is the daily sfcWindmax for day $t$ at model grid cell $i$, and $v_{98,i}$ is the 98th percentile of the sfcWindmax variable at model grid cell $i$, computed over the winter periods of the historical

period. The Sw2010 and CubEOT impact functions are shown in Fig. 3.

We apply the impact functions to our entire study domain, without calibrating the functions for specific regions or specific asset types (e.g. residential, industrial), due to limited availability of calibration data. Calibrating the impact functions at a regional, or country-specific level, or deriving impact functions for different asset types requires long records of historical loss data, at a fine spatial resolution and for different asset types. Such data was not publicly available at the time of this study.

Considering European-wide damage functions for all asset types can indeed contribute to less realistic damage estimates in absolute terms when investigating the damages at a regional or country-specific level. However, this study investigates the relative changes in the damages comparing future to historical climate conditions, hence we expect inter-regional differences in vulnerability to partly cancel out when normalizing with the historical-climate baseline. Furthermore, this approach allows a better comparison of the damages between the different regions, and is also commonly used in other similar studies (e.g.

Schwierz et al., 2010; Donat et al., 2011; Pinto et al., 2012; Meiler et al., 2023). We also assume a vulnerability to wind damage constant in time, consistently with other studies (e.g. Schwierz et al., 2010; Pinto et al., 2012; Meiler et al., 2023). Hence, vulnerability is modelled according to the historical baseline, and we do not project changes of vulnerability in time. As in the case of exposure, our assumption of constant vulnerability in time should be borne in mind when investigating results of our risk projections.

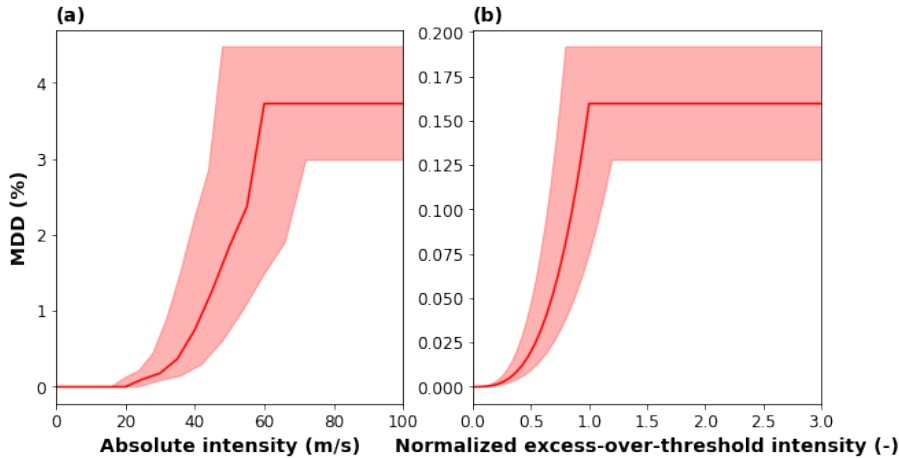

**Figure 3.** The two impact functions considered in this study. Panel (a) shows the empirical curve based on recorded winter storm losses from a reinsurance company from Schwierz et al. (2010); Panel (b) shows the Cubic excess-over-threshold from Klawa and Ulbrich (2003). For the two curves, the red bands indicate the calibration uncertainty accounted for in the uncertainty and sensitivity analysis (see Sect. 2.6).

### 2.4 Damage and risk metrics

We use primarily the following damage and risk metrics to present our damage projections:

1. *Average Annual Damage (AAD)*: The average annual damage represents the sum of all the damages occurring during a period, divided by the number of years in this period. The average annual damage is a relevant risk metric for the insurance industry, as it informs on the damages accumulated over a certain period.

2. *Exceedance Frequency Curves (EFCs)*: Exceedance frequency curves allow us to visualize the distribution of the damages in the frequency-intensity domain. Considering a damage event set covering a period of $N$ years, we can order the damage events by intensity, and assign to each damage event an exceedance frequency. For instance, the intensity of the most intense damage event being reached or exceeded only once over the whole period covered by the dataset, receives an exceedance frequency of $\frac{1}{N}$, and the intensity of the second most intense damage event reached or exceeded twice, receives an exceedance frequency of $\frac{2}{N}$, etc. Alternatively, one can consider the *Return Period (RP)* of an event, which is the inverse of the exceedance frequency, and corresponds to the time interval during which a certain event intensity is expected to not be exceeded. It has to be noted that the intensity of an event with a certain return period is based on fewer events as the return periods considered increase, thus increasing the sampling uncertainty of the statistic (Welker et al., 2021). For instance, the intensity of an event with a return period of 30 years will be based on one event only for a 30-years-long dataset, and the intensity of a 15 years return period event will be based on two events.

This study investigates the changes in damages in both the spatial, and frequency-intensity dimension. Changes in the damages in the spatial dimension are investigated using spatial maps of the change in the average annual damage, and by examining

the multi-model distribution of the changes in average annual damage, and damage amounts with return periods of one and 15 years, aggregated to a regional level. Future changes in the average annual damage, and damages for return periods of one and 15 years are presented in the form of a future-minus-historical change relative to the historical reference period. As only changes in the climate conditions are considered in this study and future changes in exposure and vulnerability are neglected, we refer to the future-minus-historical change relative to the historical reference period as *Delta Climate*. For the computation of the spatial maps resolved at the exposure level, a constant value of one is added to the historical damages, to avoid dividing by zero in regions where no damages are modelled during the historical period. We investigate frequency-intensity changes of the damages in a changing climate by comparing exceedance frequency curves of the damages aggregated to the entire study domain obtained for the future and historical climate conditions.

## 2.5 Model calibration and validation

We use WG10 from the ERA5 reanalysis regridded at $1° \times 1°$ resolution as hazard data to compute damages over a control period. We then compare the modelled damages to recorded damages retrieved from the EM-DAT database (Guha-Sapir, 2021). The damages recorded in the EM-DAT database only cover a period spanning from 1998 to 2020, and we thus use the 12 years which overlap with our historical period of 1980 to 2010 as a control period. We first ensured that the AADs computed using each impact function over the control period reproduce to a reasonable order of magnitude the AAD computed with the recorded damages of EM-DAT. The Sw2010 impact function yielded realistic damage estimates but the CubEOT impact function required calibration, as it yields damage estimates that are higher by about two orders of magnitude in comparison to the AAD of EM-DAT. We thus recalibrated the CubEOT impact function by rescaling its MDD scale by a multiplicative factor $MDD_{max}$, computed as the ratio of the AAD from EM-DAT over the ADD obtained with the uncalibrated CubEOT function during the control period:

$$MDD_{max} = \frac{AAD_{EM\text{-}DAT}}{AAD_{ERA5,control}} \tag{3}$$

Next, we controlled that both impact functions are able to produce realistic damage estimates for the ten storm events present in the EM-DAT dataset (Anatol, Calvann, Cilly, Desiree, Fanny, Emma, Erwin, Jeanett, Klaus, Kyrill, Lothar, Martin; Cilly, Desiree, and Fanny are registered as a single storm event in the EM-DAT database). For all storm events, our modelling framework underestimates the damages, resulting in a mean absolute percentage error of 78% and 66% for the Sw2010 and CubEOT impact functions respectively. However, we consider those errors as acceptable in the scope of this study, and validate our modelling framework, as it is able to produce sufficiently realistic damage estimates for major European winter storm events.

## 2.6 Uncertainty and sensitivity quantification

The uncertainty and sensitivity analysis carried out in this study is done entirely with CLIMADA's *unsequa* module and follows the essential steps described in Kropf et al. (2022). We first define input variables and input parameters that represent relevant factors of uncertainty in the modelling of winter storm damage over Europe in a changing climate. Those factors of uncertainty

relate to uncertainties in the modelling of the hazard, exposure, and vulnerability. We assess the uncertainty in the hazard data by varying the GCMs and SSPs used to generate the storm days. In addition, we assess the effect of internal variability on the hazard by considering different model members for each GCM-SSP combination. We thus select 14 GCMs that provide at least three model members for the five experiments: *historical*, *SSP126*, *SSP245*, *SSP370*, and *SSP585*. See Tab. A2 for a list of the climate models used for the uncertainty and sensitivity analysis. The choice of the climate model, future climate

scenario, and model members are reflected by the $modid$, $scenid$, and $memid$ uncertainty parameters. Each parameter is an integer index corresponding to a unique climate model, future climate scenario, or model member, and is drawn uniformly over the possible set of values for each uncertainty factor: $modid \in (0, 1, 2, ..., 13)$, $scenid \in (0, 1, 2, 3)$, and $memid \in (0, 1, 2)$. We choose uniform distributions as we assume each of the possible parameter values to have equal probability. Uncertainty in the modelling of the exposure is accounted for by varying the $m$ and $n$ exponents of the LitPop exposure data, which respectively

govern the weight given to the population count and nightlight intensity data layers used for the spatial disaggregation. Varying the $m$ and $n$ exponents allows us to simulate uncertainty in the geographical distribution of the physical assets, with higher values of $n$ emphasizing highly populated areas, and lower values of $n$ less densely populated areas (Kropf et al., 2022). According to Eberenz et al. (2020), $m = 1$, and $n = 1$ is the best performing parameterization for total value distribution in space. Hence, taking this parameterization as a basis, we generate eight additional exposure datasets combining values of $m$

and $n$ taken from a list: $m, n \in (0.75, 1, 1.25)$, assuming variations of $\pm 25\%$ in the $m$ and $n$ parameters to lead to reasonable variations in the generated exposure dataset. The parameter $f_{exp}$ represents the choice of the exposure dataset, and is uniformly drawn over nine index values: $f_{exp} \in (0, 1, 2, ..., 8)$, assuming all generated exposure layer to be equally plausible. We quantify the uncertainty associated with the functional form of the impact function by using two different impact functions to model the damages: the Sw2010 and the CubEOT impact functions (c.f., Fig. 3). Each curve represents a commonly-used approach

in the field of winter windstorm damage modelling, the former representing a more empirically risk-data driven modelling of the vulnerability, and the latter being more statistically motivated. The parameter $impfid$ represents the choice of the impact function and is drawn uniformly over the two possible index values $impfid \in (0, 1)$, as we consider both functions to be equally valid choices. Important uncertainties can also arise as a result of the calibration of an impact function. We estimate those uncertainties by perturbing the input-intensity and output-MDD scales of the impact functions with two separate

multiplicative factors $x_{scale}$ and $y_{scale}$, which we draw separately from a uniform distribution $x_{scale}, y_{scale} \in [0.80, 1.20]$. We choose boundaries for the uncertainty parameters of 0.80 and 1.20 as we estimate an error of $\pm 20\%$ to be a reasonable estimate of the error occurring in the modelling of European winter storms damages (Prahl et al., 2012). Note that we do include uncertainty in the vulnerability (two different functional forms) and the exposures (9 different urban distributions). However, we do not include uncertainty in the exposures and vulnerability projections as we do neither include changes in the exposures

nor the vulnerability for future scenarios.

We select the variance-based Sobol´ sensitivity indices as sensitivity metrics (Sobol, 2001). Using the quasi Monte Carlo sampling scheme and the computation methodology described in Saltelli et al. (2010), we generate 32768 samples and compute first order (S1), and total order (ST) Sobol´ sensitivity indices for several damage and risk metrics. First order Sobol´ sensitivity indices represent the direct contribution of a model's uncertainty factors to the model output variance. Total order Sobol´

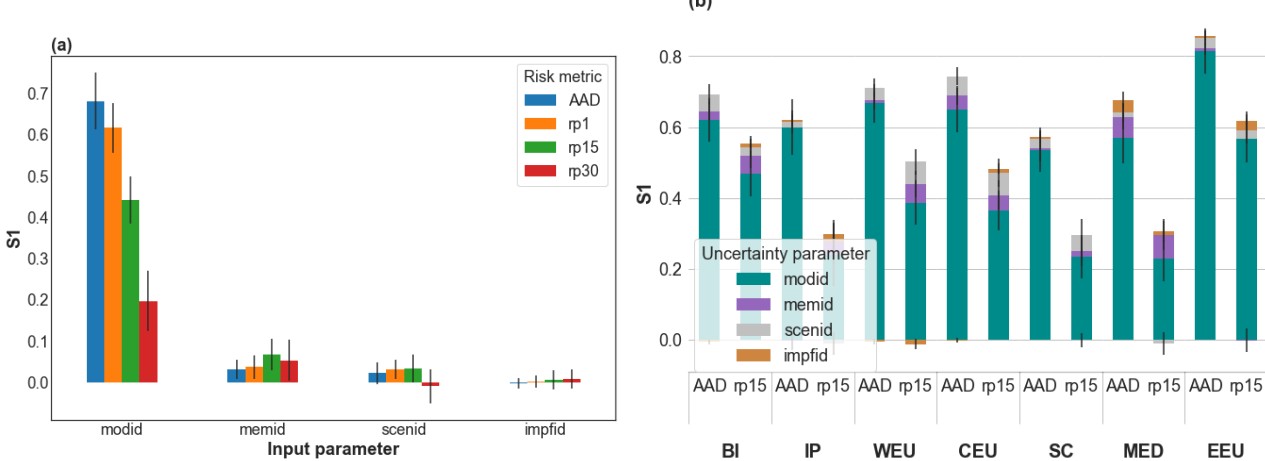

**Figure 4.** First order Sobol′ sensitivity indices (S1) for the future-minus-historical change relative to the historical period (Delta Climate) in winter storm damage in Europe, comparing a future (2070-2100) to a historical period (1980-2010). Panel **(a)** shows results for the Delta Climate in average annual damage (AAD), and in damage amounts with return periods of one, 15, and 30 years (rp1, rp15, and rp30), aggregated over all exposure points of the entire European domain; Panel **(b)** shows results for the Delta Climate in average annual damage (AAD), and in damage amount with a return periods of 15 years (rp15), aggregated over all exposure points in each of the seven regions defined in Fig. 2: British Isles (BI), Iberian Peninsula (IP), Western Europe (WEU), Central Europe (CEU), Mediterranean and Balkan region (MED), Scandinavia (SC), and Eastern Europe (EEU). The uncertainty factors (cf. sec. 2.6) are $modid$: climate model choice, $memid$: climate model member choice, $scenid$: future climate scenario choice, and $impfid$: impact function choice. The vertical black bars in **(a)** and **(b)** indicate the 95th percentile confidence intervals.

sensitivity indices represent the sensitivity to a factor including all its higher order interactions with the other uncertainty factors.

## 3   Results

We first conduct an uncertainty and sensitivity analysis to determine the dominant factors of uncertainty in the projection of winter windstorm damage in a changing climate. Secondly, we present the Delta Climate in winter windstorm damage, using the findings of the uncertainty and sensitivity analysis to restrict the analysis to the components of the projection ensemble which are the most relevant for the uncertainty in the projections.

### 3.1   Sensitivity analysis

We conduct a sensitivity analysis using Sobol´ sensitivity indices to determine the most relevant uncertainty factors for projections of the Delta Climate in winter storm damage in Europe. We determine the leading uncertainty factors at a continental scale and study the influence of considering damage events with increasingly longer return periods on the sensitivity. To do so,

we investigate sensitivity on the average annual damage, and on damage amounts with return periods of one, 15, and 30 years, aggregated over the entire domain. As a second step, we assess whether the results of the sensitivity analysis vary according to the region considered, by considering sensitivity on the average annual damage and on damage amounts with return periods of 15 years, aggregating over the seven different sub-regions defined in Fig. 2. We consider for our sensitivity analysis the Delta

Climate in the damage and risk metrics, where Delta Climate refers to the future-minus-historical changes of the metrics with respect to the historical period (c.f. Sect. 2.4).

Fig. 4a displays first order Sobol´ sensitivity indices (S1) for the Average Annual Damage (AAD), and for damages corresponding to return periods of one, 15, and 30 years (rp1, rp15, rp30), aggregated over the entire domain. Confidence intervals

obtained by bootstrapping are plotted to inform on the precision of the estimated sensitivity indices and on the convergence of the sensitivity calculation. The confidence intervals indicate that the number of samples is sufficient to reach convergence and to obtain sufficiently precise estimates of the sensitivity indices. The values of the first order sensitivity indices indicate that the choice of the climate model ($modid$) is alone responsible for almost 70% of the total variance of the projection ensemble for the average annual damage. Climate model uncertainty also accounts for about 60%, and 45% of the total variance for damage

amounts with return periods of respectively one and 15 years, but the contribution of this uncertainty factor drops to 20% for damage amounts with return periods of 30 years. The uncertainty associated with the use of different ensemble members ($memid$) ranks second, with values of the first order sensitivity indices not exceeding 10%, and the uncertainty associated with the future climate scenario ($scenid$) ranks third, with values of the first order sensitivity indices not exceeding 5%. Therefore, climate model uncertainty is the dominant uncertainty for average annual damage and damage amounts with return periods

up to 15 years when aggregating the results to the entire domain. Overall, the first order sensitivities to the impact function calibration factors $x_{scale}$, and $y_{scale}$, and to the exposure layer $f_{exp}$ are negligible for the four damage metrics here considered. Those three parameters were thus removed from Fig. 4a and b, and are ignored in the further analyses. Complete figures including the uncertainty contribution from the $x_{scale}$, $y_{scale}$, and $f_{exp}$ parameters can be found in the appendix (Fig. B1). For damages with return periods higher than 15 years, the sum of all first order sensitivity indices drops below 50%, indi-

cating that interactions between the different uncertainty factors become more important than the direct contributions of the uncertainty factors taken separately (Saltelli, 2002). We explain this increasing importance of interactions between uncertainty factors by the sampling uncertainty associated with increasing return periods. In our modelling framework, damages with return periods of 15 years and higher are based on two and one damage events respectively, which renders the estimation of the damage metrics highly influenced by sampling uncertainty. In consequence, the sensitivity analysis cannot clearly separate the

contributions of the different uncertainty factors, as those become confounded with the sampling uncertainty. Alternatively, we examine total order Sobol´ sensitivity indices (ST), as those indices inform on the total contribution of an uncertainty factor to the uncertainty, including all the interactions with the other uncertainty factors (Fig. B1a). The total order sensitivity indices also emphasize the dominance of the climate model uncertainty, as the total order sensitivity indices for the $modid$ parameter are systematically higher than the total order sensitivity indices for the other uncertainty factors. Therefore,

the climate model uncertainty remains the dominant uncertainty factor for damages with return periods of 15 years and higher.

The choice of the ensemble member ($memid$), of the future climate scenario ($scenid$), and of the impact function ($impfid$) rank respectively second, third and fourth in terms of total order sensitivity. Interestingly, the values of the total order sensitivity indices increase with the return period for the $memid$, $scenid$, and $impfid$ parameters, but slightly decrease with the return period for the $modid$ parameter. This increase of the total order sensitivity indices with the return period suggests an increasing importance of the $memid$, $scenid$, and $impfid$ uncertainty factors when investigating damages with increasing return periods.

We test the assumption that interactions between the uncertainty parameters are indeed associated with the sampling uncertainty by repeating the sensitivity analysis, but combining the three ensemble members from each climate model instead of considering the different ensemble members separately. Combining the different ensemble members of a climate model instead of considering the members separately allows us to decrease the sampling uncertainty. The results of the sensitivity analysis obtained when considering the 90 years of modelled climate instead of separate 30 year periods are shown in Fig. B2. In contrast with the results of the sensitivity analysis obtained when considering the ensemble members separately, the sum of the first order sensitivity indices increases, indicating that considering longer time periods indeed helps to decrease the interactions between the uncertainty factors. In addition, we observe that climate model uncertainty remains the dominant source of uncertainty for projections of damage amounts with return periods up to 30 years as the first order sensitivity index to the climate model is larger than 0.5 for the rp30 metric. Furthermore, the total and first order sensitivities to both the future climate scenario and impact function choice decrease when considering the combined simulations. This decrease indicates that the apparent increasing contribution of the future climate scenario and impact function uncertainty when considering damages with increasing return periods is at least partly associated with sampling uncertainty. Thus, the increasing relevance of the impact function and future climate scenario should not be over-interpreted, as this increase is likely associated with the inability of the sensitivity quantification framework to differentiate between the different uncertainty factors when assessing damages with increasing return periods.

The sensitivity analysis conducted on the average annual damage and damage amounts with a return period of 15 years aggregated to the seven regions reveals that climate model uncertainty also dominates the uncertainty at a regional scale, as the values of the first order sensitivity indices for the $modid$ parameter are systematically the highest in all regions (Fig.4b). Similarly as for the sensitivity analysis conducted for the results aggregated to the entire domain, the sums of the first order sensitivity indices decrease when damages with longer return periods (here rp15) are considered. We also note some inter-regional variations. For instance, the sensitivity to the impact function is higher in the Mediterranean region than in the other regions, when considering the average annual damage. This regional variability in the sensitivity to the impact function suggests that the design of the impact function can become an increasingly important factor of uncertainty when assessing damages at a regional level. However, we remind that this regional variability should be interpreted with care, as the effects of sampling uncertainty can be expected to increase when investigating smaller spatial scales.

Overall, our results emphasize the dominance of the climate model uncertainty when investigating the Delta Climate in winter windstorm damage over Europe. However, the sensitivity quantification framework which we use does not allow us to clearly separate between the different sources of uncertainty when the return period of the estimated damage approaches the duration

of the hazard dataset. Our findings are consistent with previous studies which also found the choice of the climate model to be a dominant factor of uncertainty for projections of winter storm damage over Europe in changing climate conditions (e.g. Leckebusch et al., 2007; Pinto et al., 2007; Donat et al., 2011; Schwierz et al., 2010). In contrast, our analysis does not show the choice of the ensemble member or the future climate scenario to be important factors of uncertainty, which is partly in contradiction with Donat et al. (2011) and Pinto et al. (2012). However, the fact that we use a different ensemble of climate models compared to these studies can potentially explain the differences in the results of the sensitivity study. In particular, we use a larger number of climate models, whose agreement on the patterns and intensities of the future changes in extreme surface winds is limited. This large ensemble of climate models probably leads our damage projections to be more uncertain than the previously mentioned studies. We can expect to find similar results with an increased relevance of the internal variability and future climate scenario by considering a less uncertain ensemble of climate models. Furthermore, we note that the results are also partly sensitive to the uncertainty and sensitivity quantification method used, as Severino (2022) found the internal variability to become the dominant source of uncertainty when considering damage amounts with return periods of eight years and higher, using similar data but using a different uncertainty and sensitivity quantification method.

We also do not find impact function uncertainty to be particularly relevant for our projections. However, we expect that considering changes in the damages in absolute terms should increase the importance of uncertainties in the impact function, as the impact function is known to be a relevant factor of uncertainty in the assessment of winter storm damage (e.g. Koks and Haer, 2020).

## 3.2 Future-climate changes in winter windstorm damage

In this section, we inspect spatial and regional patterns in the projections of the Delta Climate in winter storm damage in Europe. We first examine the results with a focus on the climate model spread, as the sensitivity analysis of the previous Sect. 3.1 highlighted the dominance of the climate model uncertainty for the damage projections. Next, we briefly investigate the influence of using different future climate scenario on the results, as we expect this aspect to be relevant for authorities, policy makers and stakeholders involved in climate change mitigation and adaptation in Europe.

### 3.2.1 Regional projections and climate model uncertainty

We focus on the climate model spread by considering a fixed impact function, Sw2010 shown in Fig. 3a, a fixed future climate scenario, SSP585, and a fixed exposure data layer obtained with LitPop's default parameterization (m=1, n=1). We use spatial maps of the average annual damage resolved at the exposure level to study the spatial patterns in the damages. We then study the multi-model agreement in the projections of several damage metrics, and investigate the variation of the multi-model agreement at a regional scale, by showing boxplots of the multi-model distributions of the average annual damage, and of damages with return periods of one, and 15 years, aggregated to the seven regions defined in Fig. 2. As before, we consider the changes in the damage metrics relative to the historical period (Delta Climate).

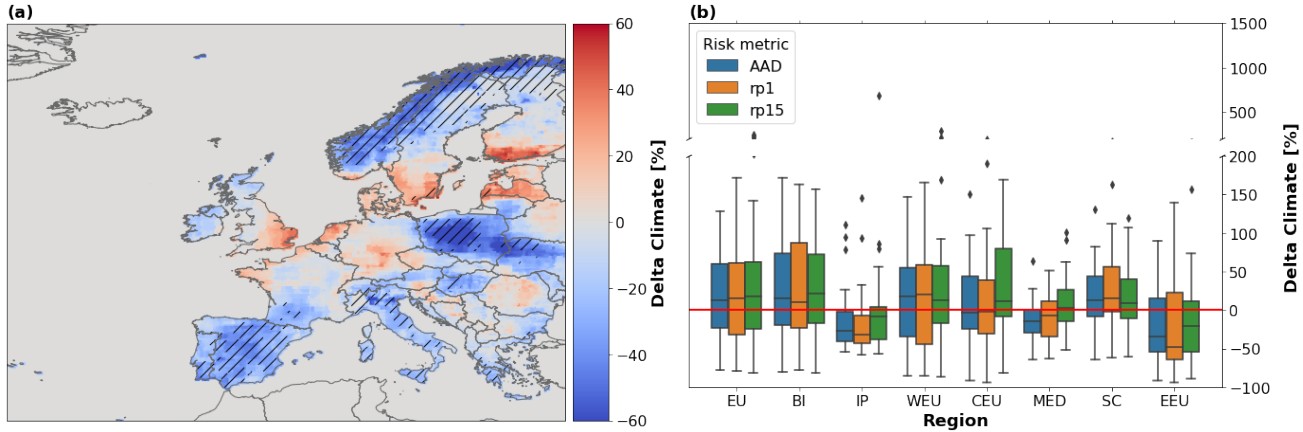

**Figure 5.** Regional changes in winter storm damage in Europe, comparing an SSP585 future (2070-2100) to a historical period (1980-2010), and using the impact function from Schwierz et al. (2010) to compute the damages. Panel **(a)** shows a spatial map of the multi-model median of the future-minus-historical change relative to the historical period (Delta Climate; %) in average annual damage, computed at each exposure point, where the hatching represents regions where more than 75% of the GCMs agree on the sign of the change in average annual damage; Panel **(b)** shows boxplots of the multi-model distributions of the future-minus-historical changes relative to the historical period (Delta Climate; %) in Average Annual Damage (AAD), and in damage amounts with return periods of one, and 15 years (rp1, rp15), aggregated over all exposure points in each of the seven regions defined in Fig. 2: British Isles (BI), Iberian Peninsula (IP), Western Europe (WEU), Central Europe (CEU), Mediterranean and Balkan region (MED), Scandinavia (SC), Eastern Europe (EEU), and over the entire European domain (EU). The boxplots' colored boxes represent the 25th and 75th percentile range (inter-quartile range) of the distributions, and the grey lines inside the boxes represent the medians. The boxplot whiskers are drawn at distances of 1.5 times the inter-quartile range (IQR) below and above the 25th and 75th percentiles of the distributions or at the minimum and maximum data points when those points fall at a distance of less than 1.5 times the IQR. The diamonds represent outlying data points outside the whiskers. The red line represents the 0-% change line, which corresponds to no change in the future-climate damages with respect to their historical value.

We first investigate the spatial pattern of the change in the Average Annual Damage (AAD), as projected by the median of the
multi-model ensemble of 30 GCMs (Fig. 5a). The multi-model median change shows increased future-climate winter storm damages within a band extending from the south of England to the Baltic states, throughout northern France, the Benelux, Denmark, Germany, and southern Sweden. In contrast, future storm damage is expected to decrease over northern Scandinavia and north-eastern Europe, as well as over the Iberian Peninsula and the Mediterranean. Parts of south-eastern Europe and the Balkan Peninsula also show a potential increase in the damages for the future period, although the signal is weaker and more
scattered. The hatching on Fig. 5a highlights regions where 75% or more of the climate models agree on the sign of the change in the average annual damage. Overall, the climate models agree well over regions where a negative Delta Climate in the damages is expected (e.g. Iberian Peninsula, Italy, Poland, and northern Scandinavia). In contrast, the climate models tend to disagree over regions where a positive Delta Climate in the damages is expected, except over limited parts of southern Sweden, and western Latvia, where the model agreement reaches or exceeds 75%.

We investigate the multi-model spread at a regional scale, by examining boxplots of the multi-model distribution of the regional changes in three damage metrics, the Average Annual Damage (AAD), and damage amounts with return periods of one and 15 years (rp1, rp15) in Fig. 5b. The boxplots' horizontal grey lines represent the medians, and the colored boxes the Inter-Quartile Ranges (IQR) of the regional distributions. The boxplots' whiskers represent either the minimum and maximum data points, or 1.5 times the IQR if the minimum and maximum data points fall beyond this distance. Data points falling beyond a distance of

1.5 times the IQR are represented by grey diamonds. Boxplots displaying the multi-model distribution of the absolute damages modelled for the historical 1980-2010 period and in the different regions are shown as reference in the appendix (Fig. B3). The boxplots in Fig. 5b confirm an important climate model uncertainty in all regions considered, with the boxplots' whiskers spanning -80% to +170% in certain regions and for certain damage metrics. Apart from the average annual damage and damage amount with a return period of one year in the Iberian Peninsula region, the 0%-change line is always covered by the

colored boxes of the boxplots, indicating that the climate model agreement on the sign of the changes is always less than 75%, at the level of the regions considered. Disregarding the important spread, the multi-model medians of the ensemble show positive changes in the average annual damage in three regions: British Isles (BI, +16%), Western Europe (WEU, +17%), and Scandinavia (SC, +13%); and negative changes in four regions: the Iberian Peninsula (IP, -28%), Central Europe (CEU, -3%), the Mediterranean and Balkan region (MED, -15%) and Eastern Europe (EEU, -35%). Aggregating the results to the entire

European domain results in an increase in the average annual damage of +13%.

We recall that even small changes in the average annual damage might result in significant damage amounts, as the damages accumulate over the years. As an illustration, we show in absolute terms the future-minus-historical difference of the accumulated damages, resulting from the accumulation of 20 years of average annual damage over the historical and the future SSP585 period. The differences between the damages accumulated over 20 years of the future period and the damages accumulated

over 20 years of the historical period yield +987 mn USD for the British Isles, +997 mn USD for Western Europe, +175 mn USD for Scandinavia, -164 mn USD for Central Europe, -45 mn USD for the Mediterranean and Balkan region, -186 mn USD for the Iberian Peninsula, and -37 mn USD for Eastern Europe, for a total of +2.7 bn USD for the entire European domain.

To better understand the spatial and regional patterns in the damages, we investigate the spatial patterns of changes for each individual climate model (not shown). While about half of the models project changes in extreme surface winds and damages

consistent with an eastward extension of the North Atlantic storm track into Europe, the remaining models project changes in extreme surface winds which cannot be straightforwardly linked to such an extension of the storm track. Furthermore, using changes in the winter-half-year-averaged monthly-mean zonal winds at 850 hPa as a proxy for changes in the storm track location, we observe a potential link between the storm track tilt, and increased damages in the Balkan. Climate models showing an intensification of damages over Northern Europe tend to be linked to a southwesterly tilted intensification of the westerly

winds, whereas models projecting damages in more southern locations are linked to a more westerly or northwesterly intensification of the westerly winds. This tilt of the low-level winds can then potentially explain the increased surface winds in the Balkans, suggesting that increased winds are a consequence of enhanced lee cyclogenesis, resulting from the blocking of the flow by the Alpine ridge. Further analysis is required in order to examine this hypothesis in more detail.

Our results are in agreement with the consensus of an eastward extension of the storm track into Europe, with numerous studies finding a similar pattern, using various data and methods (e.g., Pinto et al., 2007; Schwierz et al., 2010; Donat et al., 2011; Pinto et al., 2012; Little et al., 2023). In particular, our results are in line with the findings of Little et al. (2023) in terms of the spatial pattern and intensity of the changes. However, we find somewhat different results, in some regions, compared to other studies. For instance, Donat et al. (2011) and Pinto et al. (2012) find a pattern of damages extending over Poland, 455 whereas we find a pattern of damages extending further north, with decreased damages over Poland. We also find a weaker signal for a positive change in storm damage over the British Isles and northwestern Europe than Little et al. (2023). This difference in the results is probably partly associated with the fact that they obtain their projections of future storm damage by scaling future changes in storm severity with projected increases in population, which accounts for about 50% of the projected changes in storm damage over those regions. Another plausible explanation for the difference in the results lies in the different 460 multi-model ensembles used in the different studies. Finally, we note that the potential increase of the damages in the Balkan region has not been observed in previous studies, but that Little et al. (2023) also finds some signal for an increase in the meteorological storm severity index over this region for the SSP585 scenario.

### 3.2.2 Sensitivity to the future climate scenario

In addition, we investigate the sensitivity of the results to the future climate scenario by showing the regional boxplots of the 465 multi-model distributions (Fig. 6) and the damage maps of the multi-model median (Fig. B4) of the Delta Climate in average annual damage computed over 14 climate models using SSP126, SSP245, SSP370, and SSP585. See Tab. A2 for a list of the 14 climate models considered for the multi-scenario assessment of this section. The regional boxplots show that the multi-model distributions are partly sensitive to the future climate scenario in certain regions, as the multi-model distributions derived with scenarios corresponding to stronger future warming (SSP245, SSP370, SSP585) are shifted towards less negative changes in 470 the average annual damage when compared with a scenario of lower future warming (SSP126). This sensitivity to the future climate scenario is visible in four out of the seven regions of the domain (British Isles, Western Europe, Central Europe, and Scandinavia), and for the projections aggregated to the entire domain. In regions where the ensemble of climate models agrees well on a negative Delta Climate in the average annual damage (Iberian Peninsula, Mediterranean, and Eastern Europe), there is no marked difference between the different future climate scenarios, except in the Eastern Europe region, where projections 475 obtained using SSP585 are associated with a multi-model distribution indicating less negative Delta Climate. For the projections aggregated over the entire domain, the multi-model distributions gradually shift towards more positive Delta Climate in average annual damage, as future climate scenarios corresponding to higher warming are considered. However, this sensitivity to the warming level is less clear when investigating results at a regional level.

Using the damage maps (Fig. B4), we find the SSP126 scenario to be associated with a marked decrease in storm damage 480 over the entire domain, and a signal for increased future-climate storm damage to emerge as scenarios of higher warming are considered (SSP245, SSP370, SSP585). Our results thus indicate that the Delta Climate in storm damage partly scales to the future change in temperatures, where more moderate increases in temperature can result in larger decreases in storm damage over Europe. This increase in the damages for future climate scenario of higher warming is consistent with previous studies

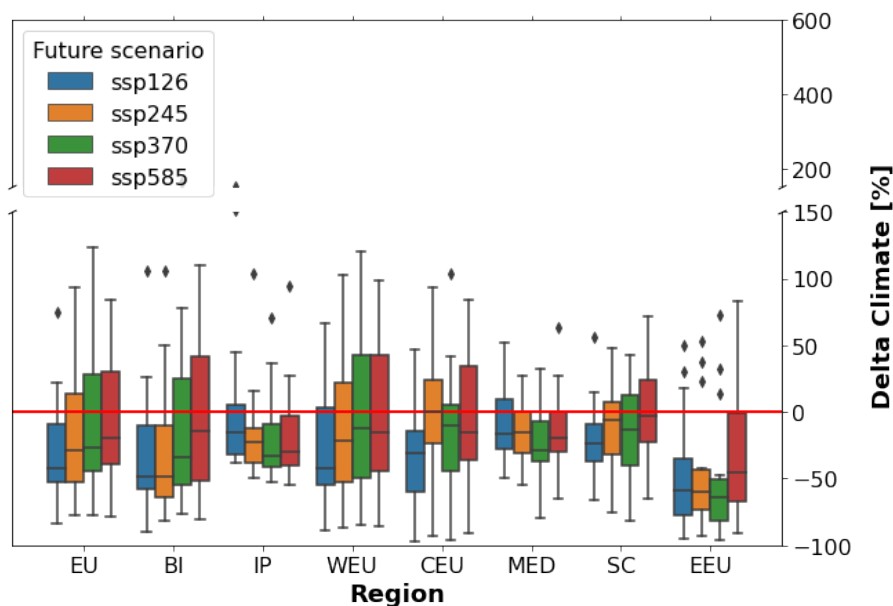

**Figure 6.** Boxplots of the multi-model distributions of the regional changes in winter storm damage in Europe, comparing different future (2070-2100) climates to a historical period (1980-2010). The boxplots show the multi-model distributions of the future-minus-historical changes relative to the historical period (Delta Climate; %) in Average Annual Damage (AAD), aggregated over all exposure points in each of the seven regions defined in Fig. 2: British Isles (BI), Iberian Peninsula (IP), Western Europe (WEU), Central Europe (CEU), Mediterranean and Balkan region (MED), Scandinavia (SC), Eastern Europe (EEU), and over the entire European domain (EU), and using four different shared socio-economic pathways (SSP126, SSP245, SSP370, SSP585) to model the future climate. The boxplots' colored boxes represent the 25th and 75th percentile range (inter-quartile range) of the distributions, and the grey lines inside the boxes represent the medians. The boxplot whiskers are drawn at distances of 1.5 times the inter-quartile range (IQR) below and above the 25th and 75th percentiles of the distributions or at the minimum and maximum data points when those points fall at a distance of less than 1.5 times the IQR. The diamonds represent outlying data points outside the whiskers. The red line represents the 0-% change line, which corresponds to no change in the future-climate damages with respect to their historical value.

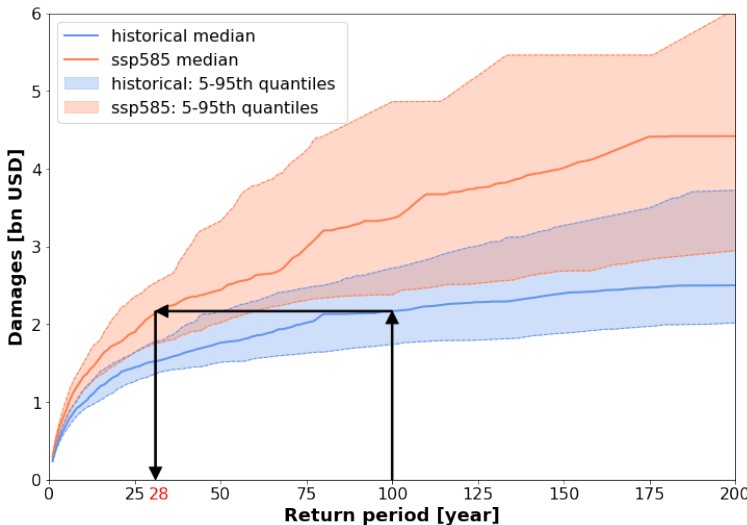

**Figure 7.** Exceedance Frequency Curves (EFCs) for future (2070-2100) SSP585 conditions, and historical (1980-2010) winter storm damage aggregated to the entire European domain. Each EFC is obtained with an ensemble of opportunity approach by combining 20 random subsamples of 20 years of data drawn from a random subselection of 20 climate models. Bootstrapped distributions of the EFCs are derived to simulate the combined effects of internal variability and climate model uncertainty. The damages are estimated using the impact function from Schwierz et al. (2010) . Solid lines represent the medians, and dashed lines the 5th and 95th percentiles of the distributions of the multiple EFCs generated via bootstrapping. The arrows highlight that the intensity of a damage event with a return period of 100 years under historical climate conditions corresponds to the intensity of a damage event with a return period of 28 years under future SSP585 climate conditions.

suggesting an increased number of cyclonic bombs over the British Isles and Western Europe for scenarios of high global warming (e.g. Zappa et al., 2013; Priestley and Catto, 2022; Little et al., 2023). However, we note that the influence of using different future climate scenarios is still limited when compared to the influence of the choice of the climate model used, as is highlighted in our sensitivity analysis (Sect. 3.1). Furthermore, we see some differences between the regional patterns of changes obtained with the set of 14 climate models used for the multi-scenario assessment, and the patterns of changes obtained with the full ensemble of 30 climate models. In particular, the multi-model median computed using 30 climate models indicates stronger regional increases in the damages, and further highlights the choice of the climate model to be the major contributor to the uncertainty in the projections. In this case, some of the climate models projecting the stronger increases in the future damages are missed when the analysis is restricted to this set of 14 climate models.

### 3.3 Ensemble of opportunity

In this section we investigate the changes in extreme damage events with long return periods under future-climate conditions by taking an ensemble of opportunity approach. Figure 7 displays the Exceedance Frequency Curves (EFCs), computed for the historical and future SSP585 periods using the ensemble of opportunity. First, we randomly sample with replacement storm

days corresponding to 20 winter seasons for each GCM and each study period (historical and future). Next, we randomly sample without replacement 20 GCMs out of the initial 30. We then combine those $20 \times 6 \times 20 = 2400$ months of data to one long simulation. Hence, each EFC represents a random realization of 2400 winter months, corresponding to 400 winter seasons. Finally, we use a bootstrapping approach, where we generate 1000 random realizations of the EFCs, and compute approximate confidence intervals using 5th and 95th percentiles of the bootstrapped distribution. This bootstrapping approach using a random subsampling on both the storm days and the climate models used to generate the EFCs allows us to estimate the combined effects of internal variability and climate model uncertainty on the projections.

The EFCs generated using the ensemble of opportunity approach reveal a considerable increase in future-climate damages with respect to historical damages. The average difference between the median EFCs of the two bootstrapped distributions obtained for the future and historical climates reveals an average increase in intensity of future-climate storm damage of 66% with respect to historical storm damage. This increase in intensity means that, for instance, damages with an expected return period of 100 years under current climate would have a return period of only 28 years under future climate conditions. The 90% confidence intervals of the historical and future curves show almost no overlap for damage amounts with return periods below 30 years. The overlap between the two confidence intervals gradually increases for damage amounts with return periods of 30 years and higher. This limited overlap between the confidence intervals of the EFCs suggests that the intensification in future-climate damages is robustly projected by the ensemble of 30 climate models for damages with return periods below 30 years, and this despite the effect of internal variability on the projections. Our analysis using the ensemble of opportunity approach thus suggests storm damage events under future 2070-2100 SSP585 climate conditions to be overall more intense than their historical counterparts. Such results are compatible with previous studies, as for instance with Schwierz et al. (2010), who found that damages with return periods of 100 years can potentially increase by 50% to 150% for Europe's end-of-century climate, and with Pinto et al. (2012), who found a potential for the frequency of extreme damage events (i.e. with return periods of 50 or 100 years) to increase by a factor of about two to four.

The investigation of the Delta Climate of the damages in the spatial dimension in the previous section revealed climate change to affect European wind storm damage with a regional heterogeneity, with some regions seeing a potential decrease in future-climate risk, and some other an increase. This section highlights a tendency of the 30 climate models considered in this study to project overall more intense winter windstorms-related damages under future climate conditions, regardless of the location of the storm damage in the study domain. Bringing together the results of the two sections thus suggests that both the location and the intensity of the windstorms will change as climate changes. This is consistent with studies projecting a decrease in the frequency of windstorms over Europe (e.g. Seneviratne et al., 2021), and studies suggesting a potential increase in the number and wind speed intensity of the most extreme extratropical cyclones over the North Atlantic and Europe (e.g. Zappa et al., 2013; Priestley and Catto, 2022). In particular, we suggest a decrease in the number of windstorm events to contribute to the decrease in future-climate average annual damage observed over some regions, and an increase in both the number and intensity of windstorm events over the western part of the domain (British Isles, northwestern France and Benelux) to contribute to an increase in future-climate average annual damage in those regions.

## 4 Summary, discussion and conclusion

In this paper, we investigate changes in winter storm damage risk over Europe in a changing climate, and assess the importance of diverse sources of uncertainty in the projections. We find large uncertainties in the projections of the Delta Climate in European winter windstorm damages, and climate model uncertainty to be the dominant factor of uncertainty in the projections. Using the median to measure the general tendency of a multi-model ensemble of 30 general circulation models, we find regional increases in winter storm damage risk over western, and northern-central Europe, and a decrease over the rest of Europe, in agreement with an eastward extension of the North Atlantic storm track into Europe. On average, we find the British Isles, Western Europe, and Scandinavia to be at risk of increased winter windstorm damage under future climate conditions, with the median of the multi-model ensemble projecting increases in average annual damage of +16%, +17%, and +13% respectively. We find a moderate decrease in the damages in Central Europe (-3%), and more marked decreases in the Iberian Peninsula (-28%), the Mediterranean (-15%) and Eastern European (-35%) regions. In order to illustrate the uncertainty in the multi-model distribution we here provide the 25th and 75th percentiles, respectively: We find changes in average annual damage of -19% and +74% for the British Isles, -35% and +55% for Western Europe, -8.9% and +44% for Scandinavia, -25% and +44% for Central Europe, -41% and -3% for the Iberian Peninsula, -30% and +1% for the Mediterranean, -55% and +15% for Eastern Europe, and -24% and +60% for the results aggregated over the entire European domain. Hence, fewer than 75% of the climate models agree on the sign of the change in all regions apart from the Iberian Peninsula. Overall, the climate model agreement on the Delta Climate in winter storm damage is poor over the regions where the damages are expected to increase according to the multi-model median, but the climate model agreement improves over the regions where the damages are expected to decrease according to the multi-model median. Using an ensemble of opportunity approach, we also find evidence for an increase in the intensity of future-climate extreme damage events, with for instance damages associated with return periods of 100 years under current climate becoming damages associated with return periods of only 28 years under future SSP585 climate conditions. Therefore, our analysis highlights climate change to induce changes in winter storm damage over Europe affecting both the spatial and the frequency-intensity domain, with different regions potentially experiencing increases or decreases in the risk of winter windstorm damage, and with overall more intense winter windstorm damage events under future climate conditions.

Overall, our results are consistent with previous studies that assessed changes in winter storm damage risk over Europe under changing climate conditions, using global or regional climate models. The spatial and regional patterns of damages that we find are in line with an eastward extension of the North Atlantic storm track into Europe in winter, consistent with findings based on previous CMIP phases (for CMIP5, see e.g. Zappa et al., 2013), and on CMIP6 (e.g. Little et al., 2023). However, we emphasize that the projections of storm damage are highly dependent on the climate model chosen, and that not all climate models show changes in extreme surface wind speeds and wind damages consistent with an eastward extension of the North Atlantic storm track. The substantial disagreement of the GCMs on the pattern of future changes of extreme surface winds over Europe in winter is not surprising, given the known difficulties of low-resolution climate models to simulate extratropical

cyclones (Seneviratne et al., 2021), and to represent regional circulation patterns and their changes (Zappa and Shepherd, 2017; Fernandez-Granja et al., 2021). Our results thus emphasize a significant disagreement on the future changes in extreme surface winds over Europe among the climate models participating in CMIP6, consistent with the findings of Kumar et al. (2015), who highlighted large inter-model differences in projections of extreme surface winds among CMIP5 climate models. Our study did not explore potential causes for the significant disagreement on the future changes in extreme surface winds we find

among the climate models. Hence, we encourage future research aimed at providing insights on this large inter-model spread. For instance, using a modelling framework similar to ours but investigating tropical cyclone damage risks, Meiler et al. (2023) found a link between the increase in damages associated with tropical cyclones and the climate model sensitivity. Exploring the link between extratropical cyclones and storm track dynamics and climate model sensitivity can provide insights into the causes of the climate model disagreement in the future change in extreme surface winds over Europe.


Addressing the shortcomings of our study, we should emphasize that our uncertainty and sensitivity analysis only covers part of the uncertainty associated with the modelling of winter windstorm damage in a future climate. Other potentially important uncertainty factors might have been missed, either because we choose not to include them (e.g. storm day detection uncertainty), or because we were not aware of them (on the concept of *unknown unknowns*, see e.g. Kundzewicz et al., 2018;

Zumwald et al., 2020). In particular, our analysis does not include future changes in exposure or vulnerability, and focuses solely on the hazard side of the risk modelling. We stress that for a complete view of future storm damage risks in Europe, changes in exposure and vulnerability should be accounted for, as those two components can be expected to have a significant influence on the outcome of the risk projection (Cremen et al., 2022). An example of a study investigating changing risks due to extreme meteorological events using a modelling framework similar to ours but including some projection in future expo-

sures can be found in Meiler et al. (2023). In this study, changing exposure is indeed shown to contribute significantly to the risk projections of tropical cyclone damage, by notably interacting non-linearly with climate change effects. Both the outputs from our risk projections and from our uncertainty-sensitivity analysis are a reflection of the uncertain input factors which we consider in our damage-risk framework, and we can expect our results to change when considering different uncertain input factors in our damage model, for instance a different number of climate models and ensemble members, additional impact

functions, a more developed modelling of the exposure and vulnerability, and different distribution and values for the uncertainty factors. The sensitivity of our results to the input data is already visible as for instance the multi-model median computed over 30 global climate models is significantly different from the multi-model median computed over an ensemble of only 14 climate models. While the high dependency of the outcome of the risk projection to the choice of the climate models reflects that this factor of uncertainty is the most relevant relative to the other factors of uncertainty examined in this study, our results

should be no means be taken as evidence that future changes in exposure and vulnerability do not matter for future changes in winter windstorm damage risk over Europe. As a final remark considering our modelling of the uncertainty, we stress that the uncertainty we quantify in this study is likely only a partial representation of the overall uncertainty, and this partial depiction of the uncertainty should be borne in mind when interpreting the confidence intervals or uncertainty estimates we derive. Regarding the climate data used to compute the damages, we found that the surface wind maxima outputs from some of

the GCMs considered in this study were subjected to significant biases when compared to the ERA5 reanalysis. We partly overcome this limitation by conducting a bias-correction based on percentile mapping, and take the approach of including all models in our analysis, despite their bias in the representation of surface winds under current and historical climate conditions. Alternatively, an approach consisting in selecting the climate models based on their performance in representing current and historical climate processes can contribute to obtain more robust projections of historical- and future-climate winter windstorm

damage. However, such a selection should be done in a way that the uncertainty associated with the climate models is still adequately represented. An example of a careful performance-based screening of climate models participating in CMIP6 can be found in Palmer et al. (2023). In addition, we expect that using an approach based on dynamical downscaling can bring significant improvements in the damage estimations (see e.g. the EURO-CORDEX project; Jacob et al., 2014). Using a dynamical downscaling approach should improve the representation of extreme surface winds over regions with complex topography, and

allow to obtain more accurate and spatially refined estimates of the future changes in extreme winds and wind damages.

    From a risk management perspective, our uncertainty and sensitivity analysis can provide valuable information for decision makers interested in the assessment of the future risks of winter windstorm damage in Europe , despite the potentially partial depiction of risks mentioned earlier. Our findings highlight the climate model uncertainty to be a key factor of uncertainty, and

we encourage actors in the risk management sector to account for this factor in their decision process. In particular, we suggest risk-friendly decision makers to base their decisions on projections from climate models which are close to the median of the multi-model ensemble, or on the median of the projections directly. On the other hand, we suggest risk-averse decision makers to base their decisions on climate models which are projecting more significant increases in winter windstorm damage under future climate conditions, by for instance selecting climate models whose projections are close to the 90th percentile of the

multi-model ensemble. To help decision makers in selecting climate model projections appropriated to their need, we rank the 30 climate models according to their projected future change in average annual damage in Tab. A1. We expect such a ranking of climate models based on results aggregated over the entire European domain to be of particular value for end-users interested in the assessment of winter storm damage risk at the scale of the entire European continent. For instance, this ranking can serve as a reference for the climate model developers community and inform the modelers on the performance of their

model regarding projections of extreme surface winds over Europe. However, for end-users interested in investigating winter storm damage risk at a more regional level, we recommend a more careful selection of the climate models, with an individual assessment of the patterns of changes in future-climate winter windstorm damage projected by the different climate models over the region of interest.

To our knowledge, our study is the first quantification of projected changes and the uncertainty of winter windstorm damage under changing climate conditions that covers such an extensive number of CMIP6 models, and which investigates uncertainty associated with modelling of the hazard, the exposure, and the vulnerability. As it stands, the large climate model disagreement in the future changes of extreme surface winds over Europe is the limiting factor in the projections of winter windstorm damage over Europe under future climate conditions. This large disagreement must be reduced in order for the impact of climate change

on future winter windstorm damage in Europe to be successfully assessed. Our findings thus motivate further research and efforts aimed at improving the representation of windstorms and the European winter climate by general circulation models. Additionally, our study emphasizes the benefits of using large ensembles of climate models for climate impact assessment studies to obtain optimal estimates of projected climate impacts. Our study also provides a common framework that can serve as guidance for future climate risk assessment studies, where a large ensemble of GCMs' projections is successfully combined with an open-source and user-friendly weather and climate risk assessment model such as CLIMADA.

*Code and data availability.* CLIMADA is openly available at GitHub https://github.com/CLIMADA-project/climada_python (last access: 23 December 2022), and https://doi.org/10.5281/zenodo.5947271 (gabrielaznar et al., 2022) under the GNU GPL license (GNU operating system, 2007). The documentation is hosted on Read the Docs https://climada-python.readthedocs.io/en/stable/ (last access: 23 December 2022) and includes a link to the interactive tutorial of CLIMADA. In this publication, CLIMADA v3.2.0, deposited on Zenodo (gabrielaznar et al., 2022) was used. Winter storm days generated from ERA5 (1980-2010) and CMIP6 models (1980-2010 and 2070-2100) used as hazard data in this study are available through the CLIMADA data API https://climada.ethz.ch/data-types/ (last access: 26 January 2024).

*Author contributions.* All authors contributed to conceptualization; data curation, formal analysis, visualization, and writing of the original draft was by Luca G. Severino; writing – review and editing – was by Luca G. Severino, Chahan M. Kropf, Hilla Afargan-Gerstman, Andries Jan de Vries, Daniela I.V. Domeisen, and David N. Bresch; supervision was by Chahan M. Kropf, Hilla Afargan-Gerstman, Christopher Fairless, Andries Jan de Vries, Daniela I.V. Domeisen, and David N. Bresch; project administration was by Daniela I.V. Domeisen and David N. Bresch.

*Competing interests.* The authors declare that they have no conflict of interest.

*Acknowledgements.* We acknowledge the World Climate Research Programme for coordinating and promoting CMIP6 through its Working Group on Coupled Modelling. We acknowledge the work of the climate modelling groups participating in CMIP6, who produced and made available their model output, and we thank the Earth System Grid Federation for archiving the data and providing access. Hilla Afargan-Gerstman acknowledges funding from the European Union's Horizon 2020 research and innovation programme under the Marie Skłodowska-Curie grant agreement No 891514. Support from the Swiss National Science Foundation through project PP00P2_198896 to Daniela I.V. Domeisen is gratefully acknowledged. This project has received funding from the European Union's Horizon 2020 research and innovation programme under grant agreements No 821010 and No 820712.

**Appendix A: Methods**

**Table A1.** Variant names, nominal resolution (Nominal res. (km)), and future-minus-historical change relative to the historical period in the average annual damage (AAD change (%)) of the different climate models considered in this study. The x symbol denotes missing ensemble member.

| Future climate scenario | Variant name | | | | | | Nominal res. | AAD change |
|---|---|---|---|---|---|---|---|---|
| | historical | | | ssp585 | | | (km) | (%) |
| Ensemble member id | 0 | 1 | 2 | 0 | 1 | 2 | | |
| Climate model | | | | | | | | |
| GFDL-CM4 | r1i1p1f1 | x | x | r1i1p1f1 | x | x | 100 | 544.6 |
| CMCC-CM2-SR5 | r1i1p1f1 | x | x | r1i1p1f1 | x | x | 100 | 217.4 |
| ACCESS-CM2 | r4i1p1f1 | r5i1p1f1 | x | r4i1p1f1 | r5i1p1f1 | x | 250 | 172.3 |
| CNRM-CM6-1-HR | r1i1p1f2 | x | x | r1i1p1f2 | x | x | 50 | 127.0 |
| CMCC-ESM2 | r1i1p1f1 | x | x | r1i1p1f1 | x | x | 100 | 78.8 |
| MPI-ESM1-2-HR | r1i1p1f1 | r2i1p1f1 | x | r1i1p1f1 | r2i1p1f1 | x | 100 | 71.3 |
| MIROC-ES2L | r1i1p1f2 | r2i1p1f2 | r3i1p1f2 | r1i1p1f2 | r2i1p1f2 | r3i1p1f2 | 500 | 68.9 |
| GISS-E2-1-G | r1i1p1f2 | x | x | r1i1p1f2 | x | x | 250 | 58.3 |
| HadGEM3-GC31-LL | r1i1p1f3 | r2i1p1f3 | r3i1p1f3 | r1i1p1f3 | r2i1p1f3 | r3i1p1f3 | 250 | 58.1 |
| MPI-ESM1-2-LR | r1i1p1f1 | r2i1p1f1 | r3i1p1f1 | r1i1p1f1 | r2i1p1f1 | r3i1p1f1 | 250 | 50.1 |
| BCC-CSM2-MR | r1i1p1f1 | x | x | r1i1p1f1 | x | x | 100 | 44.9 |
| CNRM-CM6-1 | r1i1p1f2 | r2i1p1f2 | r3i1p1f2 | r1i1p1f2 | r2i1p1f2 | r3i1p1f2 | 250 | 41.1 |
| MRI-ESM2-0 | r1i1p1f1 | r2i1p1f1 | r3i1p1f1 | r1i1p1f1 | r2i1p1f1 | r3i1p1f1 | 100 | 35.6 |
| EC-Earth3-CC | r1i1p1f1 | x | x | r1i1p1f1 | x | x | 100 | 29.7 |
| MIROC6 | r1i1p1f1 | r2i1p1f1 | r3i1p1f1 | r1i1p1f1 | r2i1p1f1 | r3i1p1f1 | 250 | 27.3 |
| IPSL-CM6A-LR | r1i1p1f1 | r2i1p1f1 | r3i1p1f1 | r1i1p1f1 | r2i1p1f1 | r3i1p1f1 | 250 | 19.7 |
| ACCESS-ESM1-5 | r1i1p1f1 | r2i1p1f1 | r3i1p1f1 | r1i1p1f1 | r2i1p1f1 | r3i1p1f1 | 250 | 5.9 |
| FGOALS-g3 | r1i1p1f1 | r3i1p1f1 | r4i1p1f1 | r1i1p1f1 | r3i1p1f1 | r4i1p1f1 | 250 | 5.8 |
| HadGEM3-GC31-MM | r1i1p1f3 | r2i1p1f3 | r3i1p1f3 | r1i1p1f3 | r2i1p1f3 | r3i1p1f3 | 100 | 4.0 |
| AWI-CM-1-1-MR | r1i1p1f1 | x | x | r1i1p1f1 | x | x | 100 | -2.6 |
| CNRM-ESM2-1 | r1i1p1f2 | r4i1p1f2 | r5i1p1f2 | r1i1p1f2 | r4i1p1f2 | r5i1p1f2 | 250 | -6.7 |
| UKESM1-0-LL | r1i1p1f2 | r2i1p1f2 | r3i1p1f2 | r1i1p1f2 | r2i1p1f2 | r3i1p1f2 | 250 | -16.5 |
| NESM3 | r1i1p1f1 | ~~r2i1p1f1~~ | x | r1i1p1f1 | ~~r2i1p1f1~~ | x | 250 | -27.2 |
| CanESM5 | r1i1p1f1 | r2i1p1f1 | r3i1p1f1 | r1i1p1f1 | r2i1p1f1 | r3i1p1f1 | 500 | -28.1 |
| EC-Earth3-Veg | r1i1p1f1 | r2i1p1f1 | r3i1p1f1 | r1i1p1f1 | r2i1p1f1 | r3i1p1f1 | 100 | -32.9 |
| EC-Earth3 | r1i1p1f1 | r3i1p1f1 | r4i1p1f1 | r1i1p1f1 | r3i1p1f1 | r4i1p1f1 | 100 | -32.9 |
| EC-Earth3-Veg-LR | r1i1p1f1 | r2i1p1f1 | r3i1p1f1 | r1i1p1f1 | r2i1p1f1 | r3i1p1f1 | 100 | -35.1 |
| INM-CM5-0 | r1i1p1f1 | x | x | r1i1p1f1 | x | x | 100 | -35.1 |
| INM-CM4-8 | r1i1p1f1 | x | x | r1i1p1f1 | x | x | 100 | -44.7 |
| KACE-1-0-G | r1i1p1f1 | r2i1p1f1 | r3i1p1f1 | r1i1p1f1 | r2i1p1f1 | r3i1p1f1 | 250 | -82.5 |

**Table A2.** Variant names of the additional ensemble members used for the uncertainty and sensitivity analysis using the SSP126, SSP245, and SSP370 experiments.

| | Variant name | | | | | | | | |
|---|---|---|---|---|---|---|---|---|---|
| Future climate scenario | | ssp126 | | | ssp245 | | | ssp370 | |
| Ensemble member id | 0 | 1 | 2 | 0 | 1 | 2 | 0 | 1 | 2 |
| Climate model | | | | | | | | | |
| CanESM5 | r1i1p1f1 | r2i1p1f1 | r3i1p1f1 | r1i1p1f1 | r2i1p1f1 | r3i1p1f1 | r1i1p1f1 | r2i1p1f1 | r3i1p1f1 |
| CNRM-CM6-1 | r1i1p1f2 | r2i1p1f2 | r3i1p1f2 | r1i1p1f2 | r2i1p1f2 | r3i1p1f2 | r1i1p1f2 | r2i1p1f2 | r3i1p1f2 |
| CNRM-ESM2-1 | r1i1p1f2 | r4i1p1f2 | r5i1p1f2 | r1i1p1f2 | r4i1p1f2 | r5i1p1f2 | r1i1p1f2 | r4i1p1f2 | r5i1p1f2 |
| EC-Earth3-Veg | r1i1p1f1 | r2i1p1f1 | r3i1p1f1 | r1i1p1f1 | r2i1p1f1 | r3i1p1f1 | r1i1p1f1 | r2i1p1f1 | r3i1p1f1 |
| EC-Earth3-Veg-LR | r1i1p1f1 | r2i1p1f1 | r3i1p1f1 | r1i1p1f1 | r2i1p1f1 | r3i1p1f1 | r1i1p1f1 | r2i1p1f1 | r3i1p1f1 |
| IPSL-CM6A-LR | r1i1p1f1 | r2i1p1f1 | r3i1p1f1 | r1i1p1f1 | r2i1p1f1 | r3i1p1f1 | r1i1p1f1 | r2i1p1f1 | r3i1p1f1 |
| MIROC-ES2L | r1i1p1f2 | r2i1p1f2 | r3i1p1f2 | r1i1p1f2 | r2i1p1f2 | r3i1p1f2 | r1i1p1f2 | r2i1p1f2 | r3i1p1f2 |
| UKESM1-0-LL | r1i1p1f2 | r2i1p1f2 | r3i1p1f2 | r1i1p1f2 | r2i1p1f2 | r3i1p1f2 | r1i1p1f2 | r2i1p1f2 | r3i1p1f2 |
| MRI-ESM2-0 | r1i1p1f1 | r2i1p1f1 | r3i1p1f1 | r1i1p1f1 | r2i1p1f1 | r3i1p1f1 | r1i1p1f1 | r2i1p1f1 | r3i1p1f1 |
| FGOALS-g3 | r1i1p1f1 | r3i1p1f1 | r4i1p1f1 | r1i1p1f1 | r3i1p1f1 | r4i1p1f1 | r1i1p1f1 | r3i1p1f1 | r4i1p1f1 |
| ACCESS-ESM1-5 | r1i1p1f1 | r2i1p1f1 | r3i1p1f1 | r1i1p1f1 | r2i1p1f1 | r3i1p1f1 | r1i1p1f1 | r2i1p1f1 | r3i1p1f1 |
| MIROC6 | r1i1p1f1 | r2i1p1f1 | r3i1p1f1 | r1i1p1f1 | r2i1p1f1 | r3i1p1f1 | r1i1p1f1 | r2i1p1f1 | r3i1p1f1 |
| MPI-ESM1-2-LR | r1i1p1f1 | r2i1p1f1 | r3i1p1f1 | r1i1p1f1 | r2i1p1f1 | r3i1p1f1 | r1i1p1f1 | r2i1p1f1 | r3i1p1f1 |
| KACE-1-0-G | r1i1p1f1 | r2i1p1f1 | r3i1p1f1 | r1i1p1f1 | r2i1p1f1 | r3i1p1f1 | r1i1p1f1 | r2i1p1f1 | r3i1p1f1 |

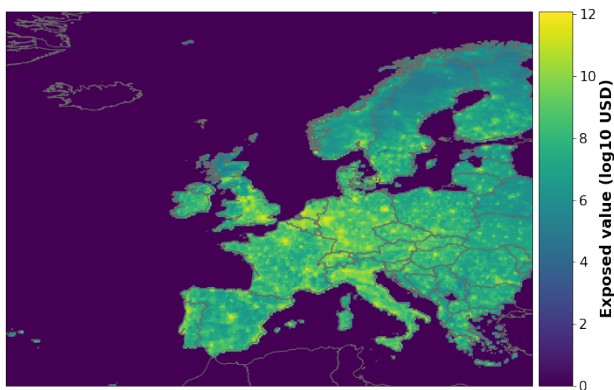

**Figure A1.** Physical assets distribution measured in USD, obtained by combining population density layer and nightlight satellite imagery, using year 2018 produced capital as the macroeconomic indicator and using the base parameterization of LitPop (cf. Litpop method Eberenz et al., 2020).

## Appendix B: Results

### B1   Uncertainty and sensitivity analysis

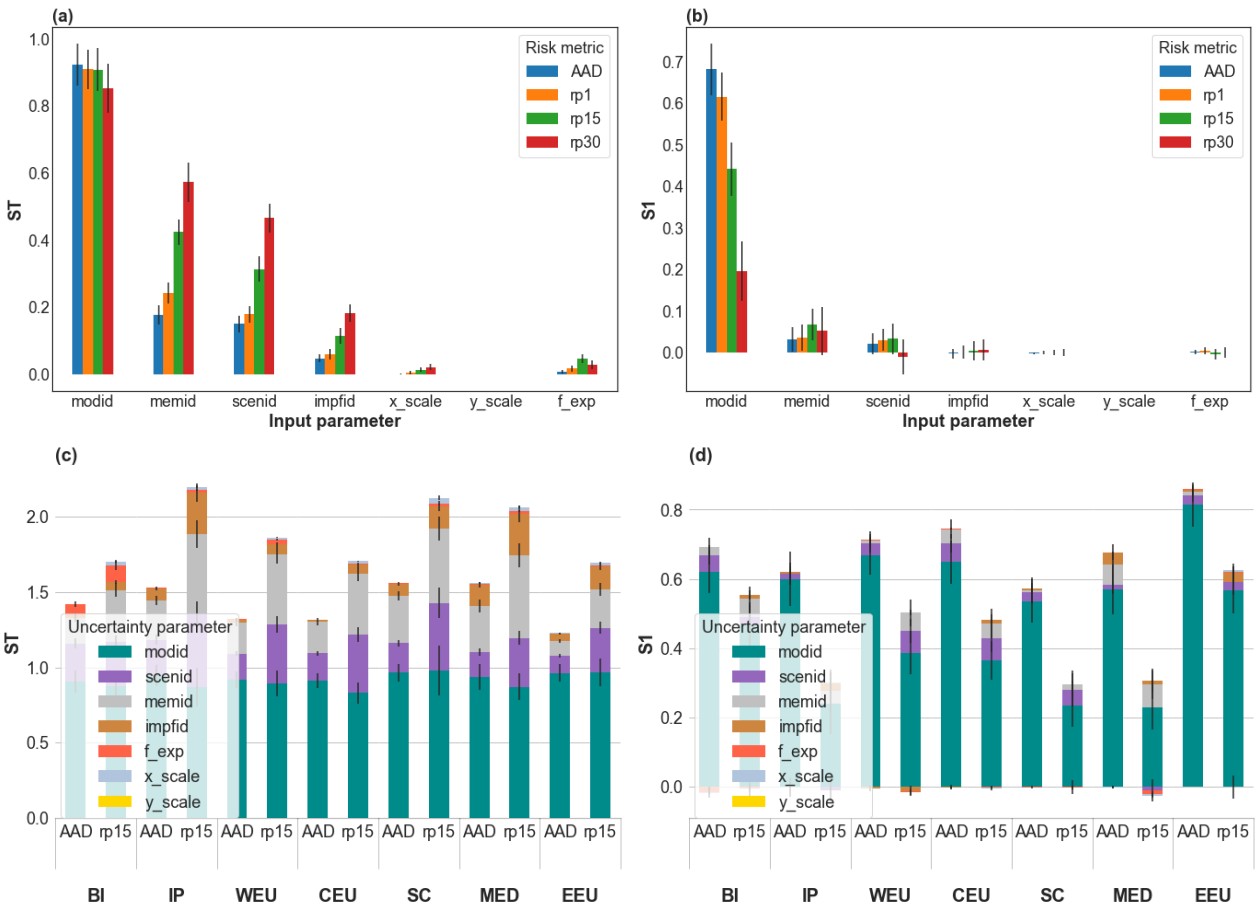

**Figure B1.** Total and first order Sobol′sensitivity indices (ST, S1) for the future-minus-historical change relative to the historical period (Delta Climate) in winter storm damage in Europe, comparing a future (2070-2100) to a historical period (1980-2010). Panels **(a)** and **(b)** respectively show ST and S1 indices for the Delta Climate in average annual damage (AAD), and in damage amounts with return periods of one, 15, and 30 years (rp1, rp15, and rp30), aggregated over all exposure points of the entire European domain; **(c)** and **(d)** respectively show ST and S1 indices for the Delta Climate in average annual damage (AAD), and in damage amount with a return periods of 15 years (rp15), aggregated over all exposure points in each of the seven regions defined in Fig. 2: British Isles (BI), Iberian Peninsula (IP), Western Europe (WEU) Central Europe (CEU), Mediterranean and Balkan region (MED), Scandinavia (SC), and Eastern Europe (EEU). The uncertainty factors (cf. sec. 2.6) are $modid$: climate model choice, $scenid$: future climate scenario choice, $memid$: climate model member choice, $impfid$: impact function choice, $x_{scale}$: impact function input intensity scaling factor, $y_{scale}$: impact function output MDD scaling factor, and $f_{exp}$: the exposure layer choice. The vertical black bars in **(a)**, **(b)**, **(c)**, and **(d)** indicate the 95th percentile confidence intervals.

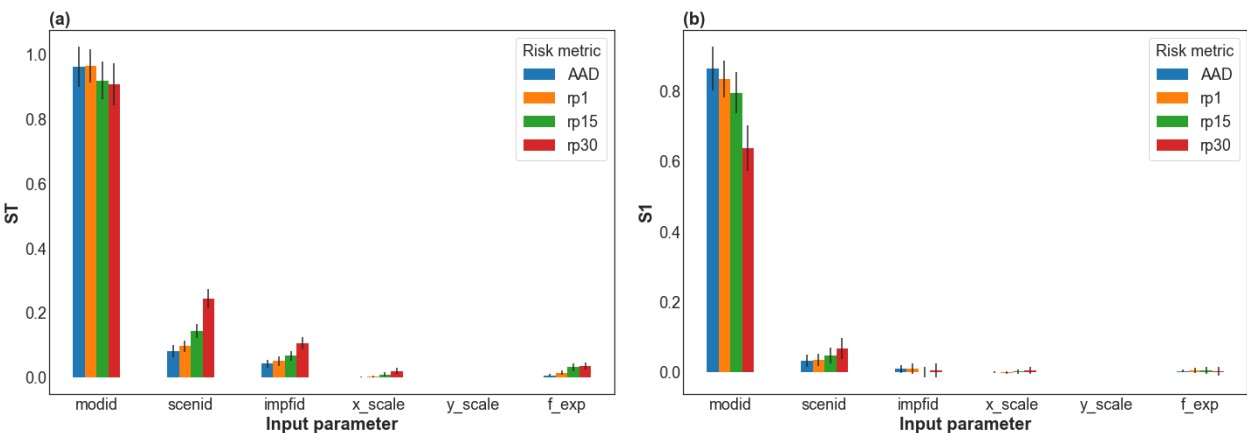

**Figure B2.** Total and first order Sobol′sensitivity indices (ST, S1) for the future-minus-historical change relative to the historical period (Delta Climate) in winter storm damage in Europe, comparing a future (2070-2100) to a historical period (1980-2010), and combining members from the climate models into single 90 years simulations. Panels **(a)** and **(b)** respectively show ST and S1 indices for the Delta Climate in average annual damage (AAD), and in damage amounts with return periods of one, 15, and 30 years (rp1, rp15, and rp30), aggregated over all exposure points of the entire European domain. The uncertainty factors (cf. sec. 2.6) are $modid$: climate model choice, $scenid$: future climate scenario choice, $impfid$: impact function choice, $x_{scale}$: impact function input intensity scaling factor, $y_{scale}$: impact function output MDD scaling factor, and $f_{exp}$: the exposure layer choice. The vertical black bars in **(a)** and **(b)** indicate the 95th percentile confidence intervals.

## B2    Future-changes changes in winter windstorm damage

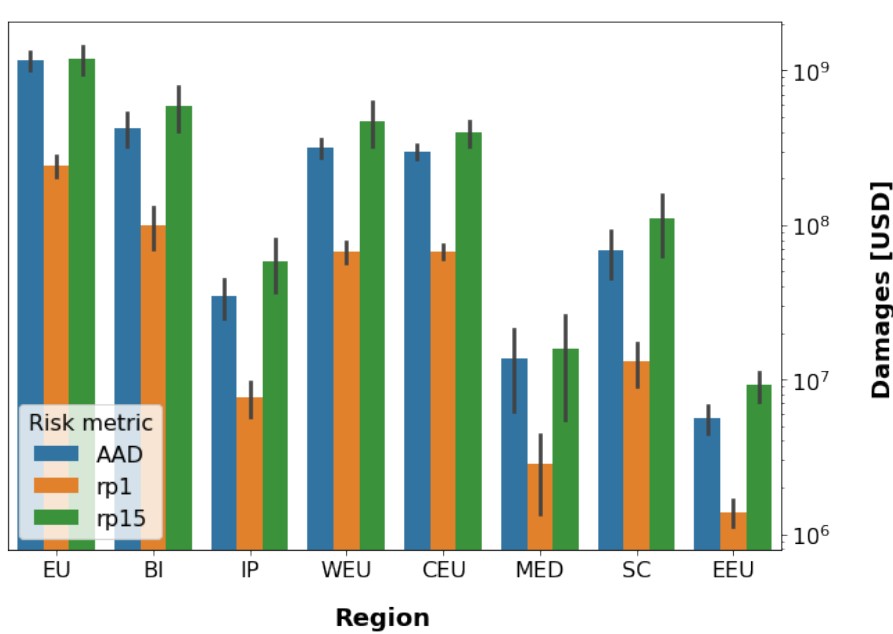

**Figure B3.** Regional projections of winter storm damage in Europe, for the reference historical period of 1980-2010. The barplots show the multi-model medians of the damages in absolute terms (USD) for the Average Annual Damage (AAD), and for damage amounts with return periods of one, and 15 years (rp1, rp15), aggregated over all exposure points in each of the seven regions defined in Fig. 2: British Isles (BI), Iberian Peninsula (IP), Western Europe (WEU), Central Europe (CEU), Mediterranean and Balkan region (MED), Scandinavia (SC), Eastern Europe (EEU), and over the entire European domain (EU). The vertical error bars represent the standard deviations of the multi-model distributions in each region.

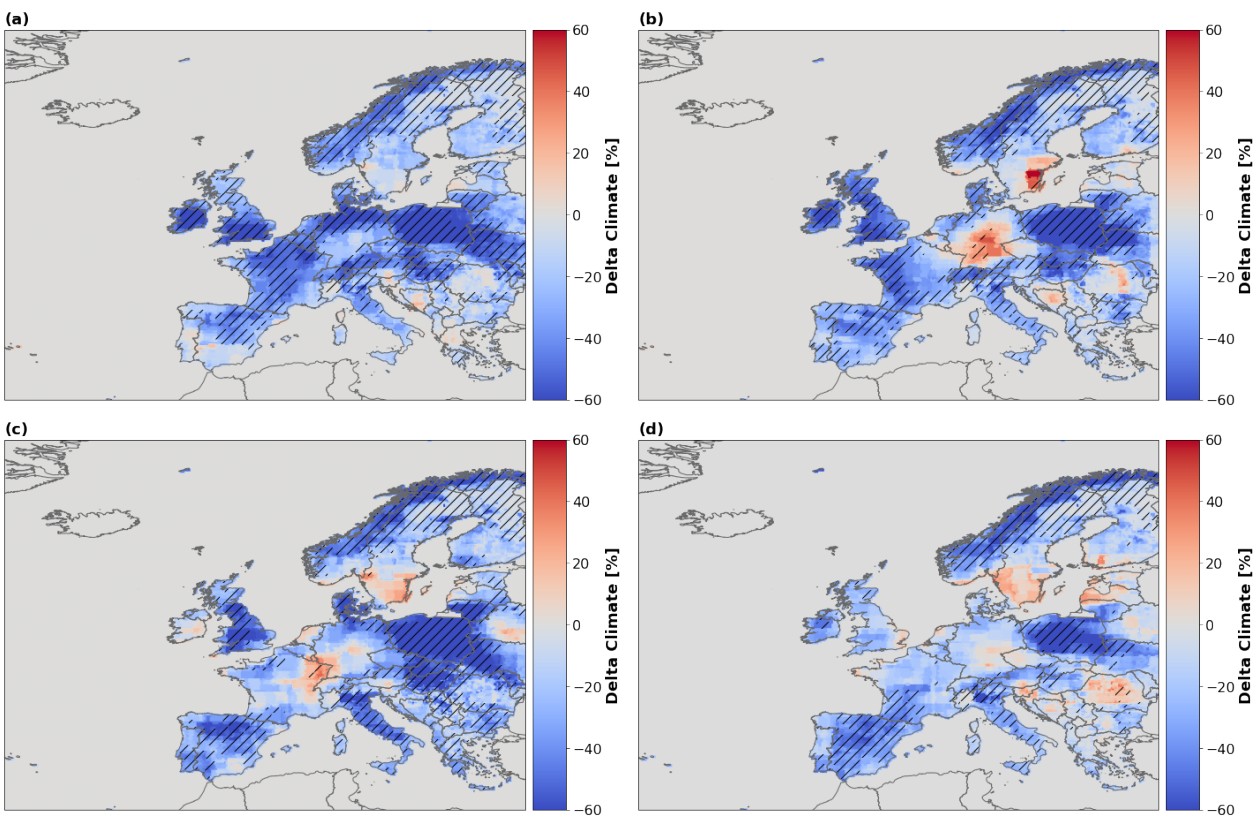

**Figure B4.** Regional changes in winter storm damage in Europe, comparing different future (2070-2100) climates to a historical period (1980-2010). Panels **(a), (b), (c),** and **(d)** show spatial maps of the multi-model median of the future-minus-historical change relative to the historical period (Delta Climate; %) in average annual damage, computed at each exposure point and using four different shared socio-economic pathways (SSP126: panel **(a)**, SSP245: panel **(b)**, SSP370: panel **(c)**, SSP585: panel **(d)**) to model the future climate. The hatching represents regions where more than 75% of the GCMs agree on the sign of the change in average annual damage.

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
