# Peer review of "Projections and uncertainties of winter windstorm damage in Europe in a changing climate"

_EGUsphere, 2023_

## Author Comment (AC1)

**Reply on referee comment 1**

Projections and uncertainties of winter windstorm damage in a changing climate

We would like to thank all reviewers for taking the time to review our work. We appreciate the insightful and helpful comments which contribute to improve our manuscript. Please find our detailed responses to the reviewers' comments and suggestions below.

The changes have been included into the manuscript and are indicated in blue here and in the annotated manuscript. In cases of corrections, phrases from the former version of the manuscript are indicated in red for reference.

The main changes to the manuscript are listed in the following. Our responses to each referee's comment are given next.

1. To account for the assumption of stationary exposure and vulnerability taken in this study, the term "future" referring to storm damage has been systematically changed to "in a changing climate", or "future-climate", or "under future climate conditions". We also replaced the term "future change", with "Delta Climate change", to clarify that the projected changes in risk refer to the changes due to future climate conditions. The title of the manuscript has also been changed accordingly to: "Projections and uncertainties of winter windstorm damage in Europe in a changing climate". Please see our reply to comment 1 of Reviewer 2 for more details.

2. Also related to point 1, clearer explanations regarding our assumptions on the modelling of the exposure and vulnerability, and their importance for the risk modelling have been included, in the description of the exposure, and vulnerability in the method section, and in the "Summary, discussion and conclusion" section. Please see our reply to comment 1 of Reviewer 2 for more details.

3. We extended our analysis by explicitly studying the effects of considering different future climate scenarios on the outcome of the damage projection. This extension has been included as an additional subsection in the results section of the manuscript. Please refer to comment 1 of Reviewer 1 for further details.

4. We found that some results in the previous version of the manuscript were erroneous as the results of the projections of the Average Annual Damage got interchanged with the results of the projections of damage events with a return period of 30 years. This modification in our results affects some details in the projections of the regional changes in the damages but does not change the main results and conclusion of the paper. To correct this inconsistency, we updated Figure 5b of the original manuscript and modified the manuscript in the results, and in the Summary, discussion and conclusion sections. The exact changes in the manuscript as well as the updated version of Figure 5b can be found in the erratum section.

**Erratum:**

Due to a mistake in the production of Fig. 5b and of some of the results presented in the body of the previous version of the manuscript, we updated Fig. 5b (see Fig. R 1 in our response below) as well as the erroneous lines in the manuscript:

[revised manuscript text omitted]

**Reviewer 1:**

**Comment 1**

The result that the scenario is not important for the uncertainty of the projections of damages is quite surprising, and as the authors point out, is not consistent with previous studies. The model variability is the largest uncertainty here. However, the use of multi-model ensembles is good because the multi-model mean will give a better representation of observations than any one realisation (e.g. IPCC 2007). There is of course a lot of uncertainty between the models, and this may be larger than the difference shown by using different SSPs. But, given the multi-model mean, it would be very informative to know the variation between the different scenarios. This is especially true given the larger changes in the storm tracks projected for higher emissions scenario (e.g. Priestley and Catto 2022). It would be good if the authors could give more information about and interpretation of this result.

**Answer:** *We thank the referee for this suggestion, and we share the referee's opinion that a more careful study of the effect of considering different future climate scenario can be insightful. Consequently, we investigate the sensitivity of the results to the future climate scenario by studying both the multi-model distribution at a regional level, and the damage maps of the multi-model median of the future change in Average Annual Damage computed over 14 climate models using SSP126, SSP245, and SSP370, and SSP585. We find our results to be partly sensitive to the choice of the climate scenario, despite the strong model uncertainty. We find this additional analysis to be a valuable complement to our study, and hence incorporated it in the body of the manuscript as new sub-subsection of the results section, entitled "Sensitivity to the future climate scenario". The new section can be found in the manuscript at lines 447-477 and is reproduced below as well as the additional figures (Fig. R 2 and Fig. R 3).*

(Lines 447-477 in the revised manuscript): In addition, we investigate the sensitivity of the results to the future climate scenario by showing the regional boxplots of the multi-model distributions (Fig. R 2) and the damage maps of the multi-model median (Fig. R 3) of the Delta Climate change in average annual damage computed over 14 climate models using SSP126, SSP245, and SSP370, and SSP585. The regional boxplots show that the multi-model distributions are partly sensitive to the future climate scenario in certain regions, as the multi-model distributions derived with scenarios corresponding to stronger future warming (SSP245, SSP370, SSP585) are shifted towards less negative changes in the average annual damage when compared with a scenario of lower future warming (SSP126). This sensitivity to the future climate scenario is visible in four out of the seven regions of the domain (British Isles, Western Europe, Central Europe, and Scandinavia), and for the projections aggregated to the entire domain. In regions where the ensemble of climate models agrees well on a negative future change in the average annual damage (Iberian Peninsula, Mediterranean, and Eastern Europe), there is no marked difference between the different future climate scenarios, except in the Eastern Europe region, where projections obtained using SSP585 are associated with a multi-model distribution indicating less negative future changes. For

the projections aggregated over the entire domain, the multi-model distributions gradually shift towards more positive Delta Climate changes in average annual damage, as future climate scenarios corresponding to higher warming are considered. However, this sensitivity to the warming level is less clear when investigating results at a regional level.

Using the damage maps (Fig. R 3), we find the SSP126 scenario to be associated with a marked decrease in storm damage over the entire domain, and a signal for increased future-climate storm damage to emerge as scenarios of higher warming are considered (SSP245, SSP370, SSP585). Our results thus indicate that Delta Climate changes in storm damage partly scale to the future change in temperatures, where more moderate increases in temperature could result in larger decreases in storm damage over Europe. This increase in the damages for future climate scenario of higher warming is consistent with previous studies suggesting an increased number of cyclonic bombs over the British Isles and Western Europe for scenarios of high global warming (e.g. Zappa et al., 2013; Priestley and Catto, 2022). However, we note that the influence of using different future climate scenarios is still limited when compared to the influence of the choice of the climate model used, as is highlighted in our sensitivity analysis (Sect. 3.1). Furthermore, we see some differences between the regional patterns of changes obtained with this set of 14 climate models, and the patterns of changes obtained with the full ensemble of 30 climate models. In particular, the multi-model median computed using 30 climate models indicates stronger regional increases in the damages, and further highlights the choice of the climate model to be the major contributor to the uncertainty in the projections. In this case, some of the climate models projecting the stronger increases in the future damages are missed when the analysis is restricted to this set of 14 climate models.

**Comment 2**

Line 140: Could the area threshold of 15000km2 be spread over wide regions? Or is there some criterion that says this is a contiguous area? I'm wondering if winds from multiple different storm features could be combined together.

**Answer:** *First, we would like to mention that the previous threshold area of 15000km2 is a typo, as the actual area threshold used for this study is 150000km2, consistently with Kruschke (2014). Secondly, different storm features at separate geographical locations can indeed be combined to trigger the detection of a storm day as no criterion on the clustering of the storm grid cells was defined. Hence, a storm day is detected as soon as the total area of the different grid cells that is identified as stormy at a particular day exceeds the threshold of 150000km2, regardless of the actual distance between the stormy grid cells. As we acknowledge that the previous formulation might lack of clarity, we incorporate this clarification in the manuscript:*

(Lines 140-142 in the revised manuscript): No clustering criterion is required on the stormy grid-cells to be included in the total stormy area required for a storm day. Hence, different wind features at separate geographical locations can be combined to evaluate whether or not the storm area ex-

[Figure]

Figure R 2: Boxplots of the multi-model distributions of the regional changes in winter storm damage in Europe, comparing different future (2070-2100) climates to a historical period (1980-2010). The boxplots show the multi-model distributions of the future-minus-historical changes relative to the historical period (Delta Climate; %) in Average Annual Damage (AAD), aggregated over all exposure points in each of the seven regions defined in Fig. 2: British Isles (BI), Iberian Peninsula (IP), Western Europe (WEU), Central Europe (CEU), Mediterranean and Balkan region (MED), Scandinavia (SC), Eastern Europe (EEU), and over the entire European domain (EU), and using four different shared socio-economic pathways (SSP126, SSP245, SSP370, SSP585) to model the future climate. The boxplots' colored boxes represent the 25th and 75th percentile range (inter-quartile range) of the distributions, and the black lines inside the boxes represent the medians. The boxplot whiskers are drawn at distances of 1.5 times the inter-quartile range (IQR) below and above the 25th and 75th percentiles of the distributions or at the minimum and maximum data points when those points fall at a distance of less than 1.5 times the IQR. The diamonds represent outlying data points outside the whiskers. The red line represents the 0-% change line, which corresponds to no change in the future damages with respect to their historical value.

ceeds the minimum threshold to count as a storm day in Europe. A value of 150000 km2 is chosen for $A_{min}$, which is representative of the typical area of the wind footprint of an extratropical storm (Kruschke, 2014).

**Comment 3**

The exposure data is interesting, but I'm curious to know how this compares to a simple population density.

**Answer:** *Thank you for the interesting comment. In general, the distributions of population and asset distributions can vary substantially, and thus the impact distribution would probably also look*

[Figure]

Figure R 3: Regional changes in winter storm damage in Europe, comparing different future (2070-2100) climates to a historical period (1980-2010). Panels **(a), (b), (c),** and **(d)** show spatial maps of the multi-model median of the future-minus-historical change relative to the historical period (Delta Climate; %) in average annual damage, computed at each exposure point and using four different shared socio-economic pathways (SSP126: panel (a), SSP245: panel (b), SSP370: panel (c), SSP585: panel (d)) to model the future climate. The hatching represents regions where more than 75% of the GCMs agree on the sign of the change in average annual damage.

*different. It would be interesting to not only study the damages to physical assets but also the number of affected people, however, this is beyond the scope of this manuscript.*

**Comment 4**

Lines 488-490: I wonder if the dynamical downscaling would actually reduce biases. Surely this depends on the large-scale/lower-resolution input? Unless the pattern is correct, but it's only the intensity that is misrepresented.

**Answer:** *We agree with this comment. What we meant is that dynamical downscaling could be a beneficial addition, especially when investigating damage projections over complex topography. We hence updated the corresponding section in the manuscript as indicated below:*

(Lines 488-490 in the previous version of the manuscript): Using a dynamical downscaling approach should improve the biases in the representation of extreme surface winds, especially over regions with complex topography, and allow to obtain more accurate and spatially refined estimates of the future changes in extreme winds and damages.

(Lines 584-588 in the revised manuscript): Using a dynamical downscaling approach should improve the representation of extreme surface winds over regions with complex topography, and allow to obtain more accurate and spatially refined estimates of the future changes in extreme winds and wind damages.

---

## Author Comment (AC2)

**Reply on referee comment 2**

Projections and uncertainties of winter windstorm damage in a changing climate

We would like to thank all reviewers for taking the time to review our work. We appreciate the insightful and helpful comments which contribute to improve our manuscript. Please find our detailed responses to the reviewers' comments and suggestions below.

The changes have been included into the manuscript and are indicated in blue here and in the annotated manuscript. In cases of corrections, phrases from the former version of the manuscript are indicated in red for reference.

The main changes to the manuscript are listed in the following. Our responses to each referee's comment are given next.

1. To account for the assumption of stationary exposure and vulnerability taken in this study, the term "future" referring to storm damage has been systematically changed to "in a changing climate", or "future-climate", or "under future climate conditions". We also replaced the term "future change", with "Delta Climate change", to clarify that the projected changes in risk refer to the changes due to future climate conditions. The title of the manuscript has also been changed accordingly to: "Projections and uncertainties of winter windstorm damage in Europe in a changing climate". Please see our reply to comment 1 of Reviewer 2 for more details.

2. Also related to point 1, clearer explanations regarding our assumptions on the modelling of the exposure and vulnerability, and their importance for the risk modelling have been included, in the description of the exposure, and vulnerability in the method section, and in the "Summary, discussion and conclusion" section. Please see our reply to comment 1 of Reviewer 2 for more details.

3. We extended our analysis by explicitly studying the effects of considering different future climate scenarios on the outcome of the damage projection. This extension has been included as an additional subsection in the results section of the manuscript. Please refer to comment 1 of Reviewer 1 for further details.

4. We found that some results in the previous version of the manuscript were erroneous as the results of the projections of the Average Annual Damage got interchanged with the results of the projections of damage events with a return period of 30 years. This modification in our results affects some details in the projections of the regional changes in the damages but does not change the main results and conclusion of the paper. To correct this inconsistency, we updated Figure 5b of the original manuscript and modified the manuscript in the results, and in the Summary, discussion and conclusion sections. The exact changes in the manuscript as well as the updated version of Figure 5b can be found in the erratum section.

**Erratum:**

Due to a mistake in the production of Fig. 5b and of some of the results presented in the body of the previous version of the manuscript, we updated Fig. 5b (see Fig. R 1 in our response below) as well as the erroneous lines in the manuscript:

[revised manuscript text omitted]

**Reviewer 2:**

**Comment 1**

It appears that the presented framework accounts only for future hazards and assumes a stationary exposure and vulnerability. In view of the current understanding of future risk (see for instance Cremen et al. 2022), this assumption appears not reasonable. If this is the case then, in my opinion, the authors are essentially quantifying the risk in a warmer climate. Can the authors clarify and modify the manuscript (and the title) accordingly?

**Answer:** *We thank the reviewer for the helpful comment. It is correct that our assumption of stationary exposure and vulnerability leads our future risk projections to reflect the effects of a changing climate only, without accounting for the effects of a changing exposure and vulnerability that can be expected to occur over time. We took this approach of time-invariant exposure and vulnerability, despite the mentioned caveats, as this allows us to focus on the hazard component, for which we dispose of a much more advanced modelling than exposure and vulnerability. This approach of exposure and vulnerability constant in time has been commonly used in the field of winter storm damage modelling (e.g. Leckebusch et al., 2007; Pinto et al., 2007; Schwierz et al., 2010; Donat et al., 2011; Pinto et al., 2012; Karremann et al., 2014). However, we agree that a complete view on future change in risk must not neglect future changes in vulnerability and exposure. On that regard, we also agree with the reviewer that a more adequate formulation can be found. We hence change the title to "Projections and uncertainties of winter windstorm damage in Europe in a changing climate", and changed all occurrences of "future" referring to the damage projections to "in a changing climate", or "future-climate", or "under future climate conditions". We also replaced the term "future change", with "Delta Climate change", to reflect that the projected changes in risk are derived considering changes in the climate conditions only. We also included clarifications in the manuscript in the description of the modelling of the exposure (lines 156-162) and vulnerability (lines 195-199), as well as for our risk metrics (lines 220-222), so that this assumption and its consequences on the outcome of the risk modelling is clearly stated. Further, we include a careful discussion of this matter in the Summary, discussion and conclusion section of our manuscript, at lines 557-563.*

(Lines 156-162 in the revised manuscript): Keeping a time-invariant exposure baseline allows us to focus on the climate change impacts on the risk outcomes and is a common approach taken in the field of natural hazard risk modelling (e.g. Leckebusch et al., 2007; Pinto et al., 2007; Schwierz et al., 2010; Donat et al., 2011; Pinto et al., 2012; Karremann et al., 2014; Meiler et al., 2023; Rana et al., 2022; Stalhandske et al., 2022). However, the assumption of a constant exposure over time yields an incomplete view of the future risk associated with storm damage, as exposure is expected to undergo considerable change over time, due to economic and societal development (IPCC, 2014). For a more complete view of future risk, changes in exposure over time should be accounted for (see e.g. Cremen et al., 2022).

(Lines 195-199 in the revised manuscript): We also assume a vulnerability to wind damage constant in time, consistently with other studies (e.g. Schwierz et al., 2010; Pinto et al., 2012; Meiler et al., 2023). Hence, vulnerability is modelled according to the historical baseline, and we do not project changes of vulnerability in time. As in the case of exposure, our assumption of constant vulnerability in time should be borne in mind when investigating results of our risk projections.

(Lines 220-222 in the revised manuscript): As only changes in the climate conditions are considered in this study and future changes in exposure and vulnerability are neglected, we refer to the future-minus-historical change as *Delta Climate* change.

(Lines 557-563 in the revised manuscript): In particular, our analysis does not include future changes in exposure or vulnerability, and focuses solely on the hazard side of the risk modelling. We stress that for a complete view of future storm damage risks in Europe, changes in exposure and vulnerability should be accounted for, as those two components can be expected to have a significant influence on the outcome of the risk projection (Cremen et al., 2022). An example of a study investigating changing risks due to extreme meteorological events using a modelling framework similar to ours but including some projection in future exposures can be found in Meiler et al. (2023). In this study, changing exposure is indeed shown to contribute significantly to the risk projections of tropical cyclone damage, by notably interacting non-linearly with climate change effects.

**Comment 2**

Lines 158-164, We only consider one single impact function for the entire domain and do not differentiate between the different asset types (e.g. residential, industrial). Considering only one single impact function allows us to avoid a complex calibration procedure requiring the tuning of numerous parameters. We use primarily one impact function, which has been designed in Schwierz et al. (2010, hereafter Sw2010), and which is already implemented in CLIMADA. This function has been directly derived from an insurer's loss model and is based on past claim data in the United Kingdom and cross-validated with other European countries. '.

Here the motivation behind the choice is apparently based on avoiding a 'complex calibration procedure'. This poses questions on the soundness of the methodology. The authors should clarify the assumptions behind their choice and its implications. Given the focus on the uncertainty, this is crucial to properly assess the robustness of the conclusions.

**Answer:** *We agree with the reviewer; this formulation is unclear, and does not allow the readers to understand the motivations for our calibration procedure. We took the approach of applying the impact functions to our entire study domain, without calibrating the functions for specific regions or specific asset types. We chose this approach for considerations of data availability, for a better regional comparability of our results, and for better comparison with existing results. Indeed, calibrating impact functions at a regional or country-specific level requires long record of damage data, which was not publicly available at the time of our study. Further, considering one impact function for the entire study domain and for all assets types allows us to readily compare the im-*

*pacts of climate change on storm damage in the different regions. However, we acknowledge that vulnerability is likely to vary regionally and that our damage projections at the regional level hence do not reflect this spatial variability. We nevertheless also note that we expect that details in the impact function have less influence when investigating the relative change in the risk, as opposed as considering absolute changes in the risk (see e.g. Meiler et al., 2023), and that this approach of using a European-wide and assets-wide damage function has already been successfully applied in previous similar studies (see e.g. Schwierz et al., 2010; Donat et al., 2011; Pinto et al., 2012). A clarification of our assumptions behind this modelling choice as well as its implication on the outcome of the risk modelling have been included in the revised manuscript:*

(Lines 186-195 in the revised manuscript): We apply the impact functions to our entire study domain, without calibrating the functions for specific regions or specific asset types (e.g. residential, industrial), due to limited availability of calibration data. Calibrating the impact functions at a regional, or country-specific level, or deriving impact functions for different asset types requires long records of historical loss data, at a fine spatial resolution and for different asset types. Such data was not publicly available at the time of this study. Considering European-wide damage functions for all asset types can indeed contribute to less realistic damage estimates in absolute terms when investigating the damages at a regional or country-specific level. However, this study investigates the relative changes in the damages comparing future to historical climate conditions, hence we expect inter-regional differences in vulnerability to partly cancel out when normalizing with the historical-climate baseline. Furthermore, this approach allows a better comparison of the damages between the different regions, and is also commonly used in other similar studies (e.g. Schwierz et al., 2010; Donat et al., 2011; Pinto et al., 2012; Meiler et al., 2023).

**Comment 3**

Lines 242, following my previous comment, here the authors state 'We quantify the uncertainty associated with the vulnerability by using two different impact functions to model the damages: the Sw2010 and the CubEOT impact functions, as this allows us to account for the uncertainty associated with the functional form of the impact function used to model the damages'. This is unclear. Is it correct that you use two functions and in both cases you assume that they apply everywhere in the European domain?' Please clarify the methodology for modelling of vulnerability and associated uncertainty.

**Answer:** *This is correct. All the winter windstorm damage projections presented in the manuscript are derived using one impact function, which we take from Schwierz et al. (2010) and which we apply to the entire study domain. As a second step, we tested the sensitivity of the results of our damage projections to the details in the damage functions by conducting an uncertainty-sensitivity analysis, where we consider an additional damage function, taken from Klawa and Ulbrich (2003).*

*However, all the results of the damage projections we present in the manuscript are derived using the impact function from Schwierz et al. (2010), as our an uncertainty-sensitivity analysis revealed the sensitivity of the damage projections to the details in the impact function to be negligible compared with the other uncertain components of our modelling framework. Hence, the uncertainty and sensitivity quantification of the modelling of the vulnerability of our risk model only accounts for the effect of varying the shape and functional forms of the impact functions. We do not assess further the uncertainty associated with the vulnerability. We modified the manuscript to clarify what is exactly accounted for in our uncertainty and sensitivity analysis of the vulnerability, at lines 267-280. Note that this allows us to compare the two main approaches to quantifying vulnerability used in literature.*

(Lines 267-280 in the revised manuscript): We quantify the uncertainty associated with the functional form of the impact function by using two different impact functions to model the damages: the Sw2010 and the CubEOT impact functions (c.f., Fig. 3). Each curve represents a commonly-used approach in the field of winter windstorm damage modelling, the former representing a more empirically risk-data driven modelling of the vulnerability, and the latter being more statistically motivated. The parameter $impfid$ represents the choice of the impact function and is drawn uniformly over the two possible index values $impfid \in (0, 1)$, as we consider both functions to be equally valid choices. Important uncertainties can also arise as a result of the calibration of an impact function. We estimate those uncertainties by perturbing the input-intensity and output-MDD scales of the impact functions with two separate multiplicative factors $x_{scale}$ and $y_{scale}$, which we draw separately from a uniform distribution $x_{scale}, \ y_{scale} \in [0.80, 1.20]$. We choose boundaries for the uncertainty parameters of 0.80 and 1.20 as we estimate an error of $\pm 20\%$ to be a reasonable estimate of the error occurring in the modelling of European winter storms damages (Prahl et al., 2012). Note that we do include uncertainty in the vulnerability (two different functional forms) and the exposures (9 different urban distributions). However, we do not include uncertainty in the exposures and vulnerability projections as we do neither include changes in the exposures nor the vulnerability for future scenarios.

**Comment 4**

I am puzzled by statements around intrinsic and epistemic uncertainty in climate projections. This is mostly due to the fact that the study relies on small ensembles that are arguably inadequate to separate the two components of the uncertainty. For instance, caption of Figure 6, 'The effects of internal variability on the EFCs are simulated by generating multiple EFCs via bootstrapping. 'Does this include resampling/subsampling over different models? The same comment applies to line 417.

**Answer:** *It is indeed difficult to disentangle the effects of intrinsic and epistemic uncertainty given the small ensemble of realizations of the climate models. The referee raises a valid point, as the fact*

*that some of the climate models only provide a single realization for a given experiment does not allow us to robustly assess the effect of internal variability for the simulations of every climate models. Please see our response to comment 7 for a more detailed explanation on how we separate the different components of uncertainty throughout our study. Regarding the results derived using the ensemble of opportunity approach, we can also expect the combined effects of climate model uncertainty and internal variability to affect the exceedance frequency curves. The bootstrapping approach introduced in the previous version of the manuscript aimed at providing an estimate of the uncertainty associated with internal variability only, repeating 1000 times a random subsampling with replacement of 20 years of the initial 30, 60 or 90 years of data available for each climate model. However, we initially did not conduct a subsampling over the climate models. In consequence to the referee's comment, we improve our bootstrapping approach by also conducting a subsampling over the different climate models. We expect that a subsampling on the different climate models can contribute to improve our estimation of the uncertainty associated with the exceedance frequency curves, as this bootstrapping allows to also assess the influence of the uncertainty associated with the choice of the climate models in the derivation the exceedance frequency curves. Consequently, we reproduce the results of Fig.6 of the initial manuscript, but including a subsampling on the climate models, so that the combined uncertainty can be assessed. See Fig. R 2 in our response for the new version of the Fig.6 of the manuscript. The subsampling over the climate models is conducted by randomly subsampling 20 climate models over the initial 30, without replacement. The results obtained using this new approach are very similar with the results obtained using the previous approach, and only a slight increase in the overlapping of the 90% confidence intervals for the different study periods can be noted. Therefore, we conclude that the projected intensification of the damages in the frequency-intensity domain is robust despite the effects of internal variability and climate model uncertainty. This enhancement in our analysis has been incorporated in the revised manuscript at lines 481-498.*

(Lines 481-498 in the revised manuscript): In this section we investigate the Delta Climate changes in extreme damage events with long return periods by taking an ensemble of opportunity approach. Figure R 2 displays the Exceedance Frequency Curves (EFCs), computed for the historical and future SSP585 periods using the ensemble of opportunity. First, we randomly sample with replacement storm days corresponding to 20 winter seasons for each GCM and each study period (historical and future). Next, we randomly sample without replacement 20 GCMs out of the initial 30. We then combine those $20 \times 6 \times 20 = 2400$ months of data to one long simulation. Hence, each EFC represents a random realization of 2400 winter months, corresponding to 400 winter seasons. Finally, we use a bootstrapping approach, where we generate 1000 random realizations of the EFCs, and compute approximate confidence intervals using 5th and 95th percentiles of the bootstrapped distribution. This bootstrapping approach using a random subsampling on both the storm days and the climate models used to generate the EFCs allows us to estimate the combined effects of internal variability and climate model uncertainty on the projections.

The EFCs generated using the ensemble of opportunity approach reveal a considerable increase in

[Figure]

Figure R 2: Exceedance Frequency Curves (EFCs) for future (2070-2100) SSP585 conditions, and historical (1980-2010) winter storm damage aggregated to the entire European domain. Each EFC is obtained with an ensemble of opportunity approach by combining 20 random subsamples of 20 years of data drawn from a random subselection of 20 climate models. Bootstrapped distributions of the EFCs are derived to simulate the combined effects of internal variability and climate model uncertainty. The damages are estimated using the impact function from Schwierz et al. (2010) . Solid lines represent the medians, and dashed lines the 5th and 95th percentiles of the distributions of the multiple EFCs generated via bootstrapping. The arrows highlight that the intensity of a damage event with a return period of 100 years under historical climate conditions corresponds to the intensity of a damage event with a return period of 28 years under future SSP585 climate conditions.

future damages with respect to historical damages. On average, future damage events are 66% more damaging than historical damage events. This increase in intensity means that, for instance, damages with an expected return period of 100 years under current climate would have a return period of only 28 years under future climate conditions. The 90% confidence intervals of the historical and future curves show almost no overlap for damage amounts with return periods below 30 years. The overlap between the two confidence intervals gradually increases for damage amounts with return periods of 30 years and higher. This limited overlap between the confidence intervals of the EFCs suggests that the intensification in future-climate damages is robustly projected by the ensemble of 30 climate models for damages with return periods below 30 years, and this despite the effect of internal variability on the projections. Our analysis using the ensemble of opportunity approach thus suggests storm damage events under future 2070-2100 climate conditions to be overall more intense than their historical counterparts.

**Comment 5**

Can you comment on the interpretation of Figs. 5 and 6. Fig.5 suggests that damage is reduced in many regions while Fig.6 indicates that overall it increases over Europe in a warmer climate. This is perfectly reasonable but more information to interpret the results could be given. A baseline value of the damage in present climate should be shown in the body of the manuscript.

**Answer:** *We thank the reviewer for this comment, and we agree that further discussion considering simultaneously the changes in the regional patterns and changes in the frequency intensity of the damages is insightful. The investigation of the Delta Climate change of the damages in the spatial dimension revealed climate change to affect European wind storm damage with a regional heterogeneity, with some regions seeing a potential decrease in future-climate risk, and some other an increase. The analysis of the changes of the damages in the intensity-frequency dimension based on the ensemble of opportunity highlights a tendency of the 30 climate models considered in this study to project overall more intense winter windstorms-related damages under future climate conditions, regardless of the location of the storm damage in the study domain. Bringing together the results of the two analyses thus suggests that both the location and the intensity of the windstorms will change as climate changes. This is consistent with studies projecting a decrease in the frequency of windstorms over Europe (e.g. Seneviratne et al., 2021), and studies suggesting a potential increase in the number and wind speed intensity of the most extreme extratropical cyclones over the North Atlantic and Europe (e.g. Zappa et al., 2013; Priestley and Catto, 2022). In particular, we can suggest a decrease in the number of windstorm events to contribute to the decrease in future-climate average annual damage observed over some regions, and an increase in both the number and intensity of windstorm events over the western part of the domain (British Isles, northwestern France and Benelux) to contribute to an increase in future-climate average annual damage in those regions. This more detailed interpretation has been added in the revised manuscript at lines 502-514.*

*Further, the suggestion of providing a baseline value of the damages in present climate is interesting, and we hence add in the appendix a barplot displaying projections of absolute average annual damages derived for the historical 1980-2010 period for the different regions (Fig. B3 in the revised manuscript and Fig. R 3 here in our response). However, we would like to stress that the focus of this study is to provide insights on the differential of storm damage risk in consequence to future changes in winter storm hazard, and that we hence consider risk metrics as future-minus-historical difference relative to the historical period. We controlled that our damage projections in absolute terms had realistic values, but these should not be interpreted as state-of-the-art estimates of winter windstorm damage under current climate conditions.*

(Lines 502-514 in the revised manuscript): The investigation of the Delta Climate change of the damages in the spatial dimension in the previous section revealed climate change to affect European wind storm damage with a regional heterogeneity, with some regions seeing a potential decrease in future-climate risk, and some other an increase. This section highlights a tendency of the 30 climate models considered in this study to project overall more intense winter windstorms-related

[Figure]

Figure R 3: Regional projections of winter storm damage in Europe, for the reference historical period of 1980-2010. The barplots show the multi-model medians of the damages in absolute terms (USD) for the Average Annual Damage (AAD), and for damage amounts with return periods of one, and 15 years (rp1, rp15), aggregated over all exposure points in each of the seven regions defined in Fig. 2: British Isles (BI), Iberian Peninsula (IP), Western Europe (WEU), Central Europe (CEU), Mediterranean and Balkan region (MED), Scandinavia (SC), Eastern Europe (EEU), and over the entire European domain (EU). The vertical error bars represent the standard deviations of the multi-model distributions in each region.

damages under future climate conditions, regardless of the location of the storm damage in the study domain. Bringing together the results of the two sections thus suggests that both the location and the intensity of the windstorms will change as climate changes. This is consistent with studies projecting a decrease in the frequency of windstorms over Europe (e.g. Seneviratne et al., 2021), and studies suggesting a potential increase in the number and wind speed intensity of the most extreme extratropical cyclones over the North Atlantic and Europe (e.g. Zappa et al., 2013; Priestley and Catto, 2022). In particular, we suggest a decrease in the number of windstorm events to contribute to the decrease in future-climate average annual damage observed over some regions, and an increase in both the number and intensity of windstorm events over the western part of the domain (British Isles, northwestern France and Benelux) to contribute to an increase in future-climate average annual damage in those regions.

**Comment 6**

Following my previous comment, how can one use the ranking provided in Tab. A1 if the signal and its robustness vary substantially from one region to another? Can you provide an example

application?

**Answer:** *It is true that the climate models strongly disagree on the patterns of changes. We expect that this ranking might be useful for users interested in the damages at the scale of the entire European continent. For instance, this ranking can serve as a reference for the climate model developers community and inform the modelers on the performance of their model regarding projections of extreme surface winds over Europe. However, for end-users interested in investigating winter storm damage risk at a more regional level, we recommend a more careful selection of the climate models, with an individual assessment of the patterns of changes in future-climate winter windstorm damage projected by the different climate models over the region of interest. This clarification has been included in the revised manuscript at lines 599-606. Furthermore, we note that the issue of selecting CMIP6 models for impact projections has been recently addressed by Palmer et al. (2023), which gives an example of a performance-based screening of different CMIP6 models with potential impact projections applications. We included this new reference at lines 580-584 in the revised manuscript.*

(Lines 580-584 in the revised manuscript): Alternatively, an approach consisting in selecting the climate models based on their performance in representing current and historical climate processes could contribute to obtain more robust projections of historical- and future-climate winter windstorm damage. However, such a selection should be done in a way that the uncertainty associated with the climate models is still adequately represented. An example of a careful performance-based screening of climate models participating in CMIP6 can be found in Palmer et al. (2023).

(Lines 599-606 in the revised manuscript): We expect such a ranking of climate models based on results aggregated over the entire European domain to be of particular value for end-users interested in the assessment of winter storm damage risk at the scale of the entire European continent. For instance, this ranking can serve as a reference for the climate model developers community and inform the modelers on the performance of their model regarding projections of extreme surface winds over Europe. However, for end-users interested in investigating winter storm damage risk at a more regional level, we recommend a more careful selection of the climate models, with an individual assessment of the patterns of changes in future-climate winter windstorm damage projected by the different climate models over the region of interest.

**Comment 7**

In section 4, the authors suggest climate model uncertainty to be the dominant factor of uncertainty in the projections. I have two points here. 1) How do you separate intrinsic and epistemic uncertainty in climate projections. 2) If indeed you have assumed a stationary exposure and vulnerability, is this conclusions essentially a consequence of your assumptions? Further comments by the authors around the interpretation of their results would be helpful .

**Answer:** *Thank you for these comments. To answer to the first point, we estimate intrinsic uncertainty by making use of hazard data generated from different realizations obtained from individual climate models. We then separate intrinsic and epistemic uncertainty in our damage projections by conducting an uncertainty-sensitivity analysis where we can compare sensitivity to the different realizations of the climate models (the internal variability) with other components making up the epistemic uncertainty in our projections (the climate model, future climate scenario, exposure layer, and impact function uncertainty). We find that climate model uncertainty largely exceeds the effects of internal variability in our projections, at least when considering average annual damage (see Fig.4 of the manuscript). For the assessment of damages amounts with return periods of 15 years and higher, we cannot successfully separate epistemic from intrinsic uncertainty, as the values of the first order sensitivity indices for the different component of the epistemic uncertainty do not sum up to a sufficient extent to explain the majority of the variance in the projections. Hence, we then present the results of our damage projections by focusing mainly on the average annual damage.We only consider damage amounts with longer return periods when taking an ensemble of opportunity approach, as this approach allows us to base the estimation of the return period events on a considerably longer time period, which we expect to significantly decrease the effects of internal variability on the projections.*

*To answer the second point, it is exact that the conclusion that we draw that climate model uncertainty is the larger contributor to the uncertainty in our projections is a direct consequence of our assumptions. We agree that this result should not foreshadow the effect of changing vulnerability and exposure on the risk outcome. As we share the reviewer's concern on this last point, we extended our discussion with a more detailed and elaborated interpretation of our results at lines 563-575. In particular, we stress that the results from our uncertainty-sensitivity analysis only reflects what is included in our modelling framework. Consequently, we cannot make statements regarding a comparison between the results of our uncertainty quantification and other uncertainties which are not accounted for in our analysis. Finally, note that we do include uncertainty in the vulnerability (two different functional forms) and the exposures (9 different urban distributions). However, we do neither include changes in the exposures nor the vulnerability for future scenarios, and consequently do not include uncertainty in the exposures and vulnerability projections.*

[revised manuscript text omitted]

---

## Author Response (AR2)

**Projections and uncertainties of winter windstorm damage in Europe in a changing climate – Authors' response 2**

The authors of the manuscript *Projections and uncertainties of winter windstorm damage in Europe in a changing climate* would like to thank once more the two anonymous reviewers and the editor for their very helpful comments and suggestions and the time they spent helping us to improve our manuscript. Please find our detailed responses to the reviewers' comments and suggestions below.

The changes have been included into the manuscript (indicated in blue in the annotated manuscript). All line indications refer to the new (annotated) version of the manuscript.

**Reviewer 1:**

**Comment 1:**
*I noticed that the authors included a reference to Little et al 2023 in the introduction, which is a very relevant recent paper. It would be good if they could give a brief sentence about how their results compare with the results of that study, particularly at the end of the results section where other studies are compared and contrasted. There seem to be some differences in the projections over northwestern Europe, which could be associated with the different models used.*

**Response:**
Thank you very much for this helpful suggestion. We added a more detailed comparison of the results from the two studies at the end of the section *3.2.1 Regional projections and climate model uncertainty* (lines 452-462 in the revised manuscript):

In particular, our results are in line with the findings of Little et al. (2023) in terms of the spatial pattern and intensity of the changes. However, we find somewhat different results, in some regions, compared to other studies. For instance, Donat et al. (2011) and Pinto et al. (2012) find a pattern of damages extending over Poland, whereas we find a pattern of damages extending further north, with decreased damages over Poland. We also find a weaker signal for a positive change in storm damage over the British Isles and northwestern Europe than Little et al. (2023). This difference in the results is probably partly associated with the fact that they obtain their projections of future storm damage by scaling future changes in storm severity with projected increases in population, which accounts for about 50% of the projected changes in storm damage over those regions. Another plausible explanation for the difference in the results lies in the different multi-model ensembles used in the different studies. Finally, we note that the potential increase of the damages in the Balkan region has not been observed in previous studies, but that Little et al. (2023) also finds some signal for an increase in the meteorological storm severity index over this region for the SSP585 scenario.

**Reviewer 2:**

**Comment 1:**
*The schematic in Figure 1 could be improved. There are two panels labelled with the letter a) which is not used in the caption.*

**Response:**
Thank you for this remark. We removed the unused captions from Figure 1 and updated the figure on the revised manuscript.

**Comment 2:**
*I find the concept of Delta Climate useful to communicate the results of the study to a broad audience, however, its definition should be provided early in the manuscript (note that the term is used also in Figure 1) and more details on the definition should be provided.*

**Response:**
Thank you for the suggestion. We added an extra clarification on the term Delta Climate at the beginning of the section *2 Data & Methods* (lines 97-101 in the revised manuscript):

Climate change effects are studied by comparing damages computed for a future (2070-2100) versus a historical (1980-2010) period while keeping exposure and vulnerability invariant in time. We present our results as the difference between the damages computed for the future and the damages computed for the historical reference period, divided by the damages computed for the historical reference period. We call this approach Delta Climate, as it informs on the change in winter storm damage associated with changing climate conditions but disregarding future changes in exposure and vulnerability.

Additionally, a detailed definition of the Delta Climate is given in the caption of Figure 1.

**Comment 3:**
*Line 491, the statement 'On average, damage events under future climate conditions are 66% more damaging than historical damage events' is not clear, it appears that it refers to ECFs shown in Fig.7 and not to simulated events.*

**Response:**
Many thanks for this helpful comment. We changed the formulation to help clarify this part of the results section (lines 505-507 in the revised manuscript):

The average difference between the median EFCs of the two bootstrapped distributions obtained for the future and historical climates reveals an average increase in intensity of future-climate storm damage of 66% with respect to historical storm damage.

**Comment 4:**
*Line 522, 'In particular, we find..', here you put forward some numbers in your conclusions but the associated uncertainty is mentioned but not discussed quantitatively. Can you clarify what are the implications of the uncertainty for these numbers?*

**Response:**
Thank you for the suggestion. We added an extra sentence to clarify the implications of the uncertainty for these numbers in the *Summary, discussion, and conclusion* section (lines 541-546 in the revised manuscript):

In order to illustrate the uncertainty in the multi-model distribution we here provide the 25th and 75th percentiles, respectively: We find changes in average annual damage of -19% and +74% for the British Isles, -35% and +55% for Western Europe, -8.9% and +44% for Scandinavia, -25% and +44% for Central Europe, -41% and -3% for the Iberian Peninsula, -30% and +1% for the Mediterranean, -55% and +15% for Eastern Europe, and -24% and +60% for the results aggregated over the entire European domain. Hence, fewer than 75% of the climate models agree on the sign of the change in all regions apart from the Iberian Peninsula.